# MANIFOLD INDUCED BIASES FOR ZERO-SHOT AND FEW-SHOT DETECTION OF GENERATED IMAGES

**Jonathan Brokman**[*†‡]
jonathanbrok@gmail.com

**Amit Giloni**[*‡]
amit.giloni@fujitsu.com

**Omer Hofman**[*‡]
omer.hofman@fujitsu.com

**Roman Vainshtein**[‡]
roman.vainshtein@fujitsu.com

**Hisashi Kojima**[⌗]
hisashi.kojima@fujitsu.com

**Guy Gilboa**[†]
guy.gilboa@ee.technion.ac.il

## ABSTRACT

Distinguishing between real and AI-generated images, commonly referred to as 'image detection', presents a timely and significant challenge. Despite extensive research in the (semi-)supervised regime, zero-shot and few-shot solutions have only recently emerged as promising alternatives. Their main advantage is in alleviating the ongoing data maintenance, which quickly becomes outdated due to advances in generative technologies. We identify two main gaps: (1) a lack of theoretical grounding for the methods, and (2) significant room for performance improvements in zero-shot and few-shot regimes. Our approach is founded on understanding and quantifying the biases inherent in generated content, where we use these quantities as criteria for characterizing generated images. Specifically, we explore the biases of the implicit probability manifold, captured by a pre-trained diffusion model. Through score-function analysis, we approximate the curvature, gradient, and bias towards points on the probability manifold, establishing criteria for detection in the zero-shot regime. We further extend our contribution to the few-shot setting by employing a mixture-of-experts methodology. Empirical results across 20 generative models demonstrate that our method outperforms current approaches in both zero-shot and few-shot settings. This work advances the theoretical understanding and practical usage of generated content biases through the lens of manifold analysis.

## 1 INTRODUCTION

Advancements in generative models, particularly diffusion-based techniques, have resulted in the creation of synthesized images that are increasingly difficult to distinguish from authentic ones. This poses significant challenges in content verification, security and combating disinformation, driving the demand for reliable mechanisms to detect AI-generated images.

A wide array of contemporary research has focused on this task, employing methods ranging from standard convolutional neural networks (CNNs) (Wang et al., 2020; Baraheem & Nguyen, 2023; Epstein et al., 2023; Bird & Lotfi, 2024) to approaches that distinguish hand-crafted and learned characteristics (Bammey, 2023; Martin-Rodriguez et al., 2023; Zhong et al., 2023; Wang et al., 2023; Tan et al., 2024; Chen et al., 2024). Despite these efforts, there is a consensus on the critical importance of generalization to unseen generative techniques in this field (Bontcheva et al., 2024): These techniques evolve quickly, presenting substantial challenges in maintaining up-to-date generated datasets, which are crucial for supervised detection methods. Methods that generalize to unseen generative techniques reduce the need for constant data collection and retraining.

Targeted efforts to enhance such generalization have been actively pursued in a specific semi-supervised setting, where models are trained on one generative technique and evaluated on another (Ojha et al., 2023; Sha et al., 2023). Notably, these methods still require data consisting of hundreds of thousands of diverse generated images. In recent months, zero-shot and few-shot techniques have emerged for this task (Cozzolino et al., 2024a; Ricker et al., 2024; He et al., 2024; Cozzolino et al., 2024b). Zero-shot methods use pre-trained models to solve tasks they were not trained for without designated (or any additional) training. Few-shot methods employ a similar tactic but involve

[†]Technion - Israel Institute of Technology [‡]Fujitsu Research of Europe [⌗] Fujitsu Limited [*]Equal contribution.

minimal data to adapt the pre-trained model (e.g. incorporating a lightweight classifier). While these methods hold great potential by eliminating the need for extensive training and data maintenance, there remains significant room for improvement in terms of operating on a wide array of (unseen) generative techniques. Moreover, current methods lack theoretical grounding - a surprising gap given the success of Mitchell et al. (2023), which revolutionized zero-shot generated **text detection** using a theoretically grounded curvature criterion. They rely on the hypothesis that generated data lies on regions of local maxima of the learned data probability. However, their method uses the explicit probability modeling of LLMs, which is impractical when addressing the typically implicit approaches of image generative models.

This work advances solutions to these gaps. It is grounded in theoretical understanding - capitalizing on the diffusion model's implicitly learned probability manifold $p$ to quantify inherent biases of its generated outputs. Namely, we present novel derivations aimed at quantifying the stability of output images along the diffusion models' generation process, yielding a new stability criterion. By design, these stable points correspond to images the model is biased to produce. To this end, we leverage the ability of pre-trained diffusion models to approximate the score function, expressed as

$$S = \nabla \log p, \tag{1}$$

and explore novel ways of analyzing the manifold defined by a function surface $\log p$. We consider several criteria, such as

$$H \propto \nabla \cdot \left( \frac{\nabla \log p}{|\nabla \log p|} \right), \tag{2}$$

the (hyper) surface curvature. This curvature is infact a total-variation differential, and is explained below as the local flux of $\frac{\nabla \log p}{|\nabla \log p|}$ [1]. Note that unlike $S$, it is not straightforward to access $H$. In our derivations, $H$ emerges from expressing the similarity between signal and noise predictions alongside two additional quantities - the gradient magnitude and a statistical bias-based quantity - which capture generation biases as well. We develop mathematically-grounded methods to access these quantities and introduce a pioneering zero-shot diffusion model analysis for detecting generated images (Fig. 1). Extended capabilities to the few-shot regime are provided as well.

**Key Contributions:**

- We establish a theoretical framework by integrating manifold analysis with diffusion model score functions, introducing a novel, bias-driven criterion for distinguishing real and generated images. This sets the foundation for further theoretical investigations in the domain.

- We propose the first zero-shot analysis of pre-trained diffusion models for generated image detection. Remarkably - this analysis demonstrates excellent generalization to unseen generative techniques.

- Our method demonstrates superior performance over existing approaches in both zero-shot and few-shot settings, validated through comprehensive experiments on a diverse dataset of approximately 200,000 images across 20 generative models.

To reproduce our results, see our official implementation[2].

## 2 RELATED WORKS

The evolution of detecting AI-generated images has primarily relied on supervised learning methodologies. The common approach utilizes standard CNNs trained on labelled datasets of real and generated images Wang et al. (2020); Baraheem & Nguyen (2023); Epstein et al. (2023); Bird & Lotfi (2024). Subsequent research by Bammey (2023); Martin-Rodriguez et al. (2023); Zhong et al. (2023); Wang et al. (2023); Tan et al. (2024); Chen et al. (2024) identified and integrated key phenomenological features, enhancing performance. These relied on extensive generated image datasets from various generative techniques, limiting their generalizability to unseen techniques Ojha et al. (2023).

---

[1]Various high-dimensional curvature definitions exist, this is our choice. Note that the TV-curvature connection itself is not new - see e.g. the mean curvature flow (MCF) perspective Kimmel et al. (1997); Aubert et al. (2006), and its generalization in Brokman & Gilboa (2021); Brokman et al. (2024)

[2]https://github.com/JonathanBrok/Manifold-Induced-Biases-for-Zero-shot-and-Few-shot-Detection-of-Generated-Images

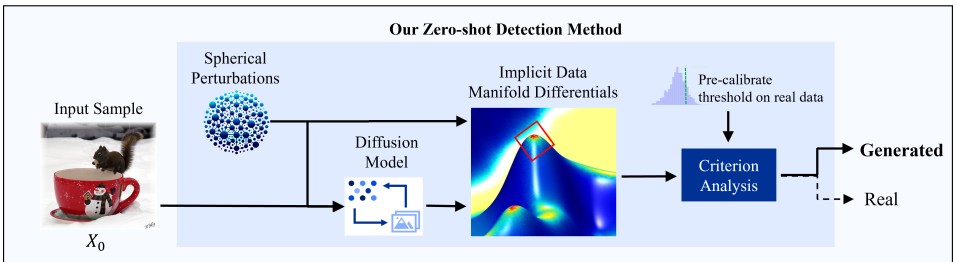

Figure 1: The proposed zero-shot detection pipeline, which circumvents the need for generated data. An input image $x_0$ is subjected to a pre-trained diffusion model and spherical perturbations. This sets the stage for our mathematical characterization of $x_0$, resulting in a criterion for the detection task.

Recent studies proposed alternative methods to enhance generalization to unseen generative techniques. Those unsupervised and semi-supervised methods Zhang et al. (2022a;b); Qiao et al. (2024); Cioni et al. (2024) reduce the reliance on extensive labeled datasets; however, they still rely on access to generative methods during training, leading to biases towards those generation techniques. Ojha et al. (2023) and Sha et al. (2023) introduced a notable CLIP-based approach to analyze content from a limited set of generative techniques, achieving unmatched generalization to unseen methods. Cozzolino et al. (2024a) extends this approach and capitalizes on generalization in the low-data regime. Using few-shot analysis with pre-trained CLIP, they outperform existing methods.

To eliminate data maintenance and training altogether, zero-shot approaches emerged. To the best of our knowledge, Ricker et al. (2024) and He et al. (2024) are the only such methods. The former employed a pre-trained auto-encoder (AE) for out-of-distribution analysis - a well-established technique, e.g. An & Cho (2015). Pre-trained on real images, the AE is expected to encode-decode them better than generated images. Therefore, the reconstruction error is their criterion for detection. He et al. (2024) compares the image representation similarity between an image and its noise-perturbed counterpart, offering insights into the desired qualities representations for the detection task. Our proposed method is the first to analyze diffusion models in the zero-shot setting for the detection task, outperforming both Ricker et al. (2024) and He et al. (2024) in terms of operating on a wide array of unseen generative techniques.

## 3 PRELIMINARIES

### 3.1 DIFFUSION MODEL SETTING

Diffusion models are generative models that operate by an iterative generation process of noise reduction based on a pre-set noise schedule. Let the data manifold be $\Omega \subset \mathbb{R}^d$, where $d$ is the data dimensionality, and denote a sample $x \in \Omega$. Each iteration $t$ involves denoising a noised signal $x_t$ via a neural network $f(x_t, t; \theta)$ ($\theta$ are the tuneable weights), subsequently progressing to $x_{t-1}$. This sequence begins at $t = T$ by sampling a tensor of iid normally distributed entries i.e. $x_T \sim \mathcal{N}(0, I)$, and terminates at $x_0 \in \Omega$, representing the final output. In our setting $x_0$ is an image. This generation process is known as *reverse diffusion*, where during training, $f$ is optimized to reverse a *forward diffusion* process, defined using scheduling parameter $\alpha_t \in \mathbb{R}^+ \ \forall t$, and noise $\epsilon \sim \mathcal{N}(0, I)$ as follows

$$x_t = \sqrt{1 - \alpha_t} x_0 + \sqrt{\alpha_t} \epsilon. \tag{3}$$

### 3.2 SCORE-FUNCTION IN DIFFUSION MODELS

The score-function is defined as $\nabla \log p(x)$, where $p(x)$ is the probability of $x$. Let $p_{\alpha_t}(x_t)$ be the probability of $x_t$ considering Equation (3). The founding works of today's diffusion models Song & Ermon (2019; 2020); Kadkhodaie & Simoncelli (2021) capitalize on Equation (3), analyzing it from a score-function perspective based on the following seminal result by Miyasawa et al. (1961)

$$\nabla \log p_{\alpha_t}(x_t) = \frac{1}{\alpha_t} \left( \sqrt{1 - \alpha_t} \mathbf{E}_x[x_0|x_t] - x_t \right), \tag{4}$$

where $\mathbf{E}_x[x_0|x_t]$ is the Minimum Mean-Squared-Error (MMSE) denoiser of $x_t$. It is replaced with the output of a denoising model $\hat{x}_0 = f(x_t, t; \theta)$, for which

$$\nabla \log p_{\alpha_t}(x_t) \approx \frac{1}{\alpha_t} \left( \sqrt{1 - \alpha_t} f(x_t, t; \theta) - x_t \right). \tag{5}$$

Often, $f(x_t, t; \theta)$ predicts noise, i.e. $\hat{x}_0 = \frac{1}{\sqrt{1-\alpha_t}} \left( x_t - \sqrt{\alpha_t} f(x_t, t; \theta) \right)$. Finally, replacing $f(x_t, t; \theta)$ with the true $x_0$ we have $\nabla \log p_{\alpha_t}(x) = -\frac{1}{\sqrt{\alpha_t}} \epsilon$, i.e. Equation (5) approximates $\epsilon$ up to a known factor. It is straightforward that, despite $\Omega$ being a zero-measure of $\mathbb{R}^d$ ($\Omega$ is assumed to have a dimension much lower than $d$), the probability of $x_t$ is non-zero on the entire $\mathbb{R}^d$ space.

With $\nabla \log p$, the generation process can be simulated as Itô's SDE Ito et al. (1951),

$$\dot{x}(\tau) = \nabla_x \log p(x(\tau)) + \sqrt{2} \mathbf{w}_\tau, \tag{6}$$

where $\dot{x}(\tau)$ is the time derivative of $x(\tau)$, and $\mathbf{w}_\tau$ the time derivative of Brownian motion $\mathbf{w}(\tau)$, i.e. it injects noise to the process. In Song & Ermon (2019), a generative process that accounts for a $p_{\alpha_t}$ that changes in time was introduced - generalizing Equation (6). With that said, this paper employs a fixed-point analysis of $p_{\alpha_t}$ for a fixed $\alpha_t$. Remark: The commonly used time here is reversed, i.e. large $\tau$ implies small $t$. The consensus in diffusion models research and this paper is to reliably use $t$.

## 4 METHOD

Here we present our mathematical perspective. For illustrations see Figs. 3, 4(b-c).

### 4.1 KEY QUANTITIES AND CRITERIONS IN OUR FIXED-POINT GEOMETRIC ANALYSIS

In the setting of Sec. 3, following Equation (6) (or its annealed version), a generated sample $x_0$ is expected to be near a stable local maximum in the learned log probability manifold - namely a point of positive curvature and low gradient - i.e. the learned manifold is expected to be "bumpy", with generated data appearing near peaks. Conversely, real data points that are unlikely generated will not exhibit these characteristics. For a graphical illustration, and a demonstration, see Figs. 2, 3(a). This aligns with Mitchell et al. (2023)'s hypothesis on generated text. To test $x_0$ for these characteristics, we work in its local neighbourhood. Let us employ a fixed-point analysis, "freezing" the generative process at a small fixed $t$. We assume $\alpha_t$ is small enough for $p_{\alpha_t}$ to approximate the data distribution and large enough for $p_{\alpha_t}$ to be smooth. Since $t$ is fixed, we use $\alpha$ from now on. Relying $p_\alpha$'s smoothness, we use $\log p_\alpha$ to construct a $d$-manifold in $\mathbb{R}^{d+1}$ as a parametric hyper-surface of the form $(x, \log p_\alpha(x))$, for which the total-variation curvature of Equation (2) applies.

Let $B_0$ be the local neighbourhood of $x_0$ with boundary $\partial B_0$, and denote their respective volumes as $|B_0|, |\partial B_0|$. Let $\langle \cdot, \cdot \rangle, \| \cdot \|_2$ denote the Euclidean inner product and norm. A gradient criterion

$$D(x_0) := \frac{1}{|\partial B_0|} \int_{\partial B_0} \| \nabla \log p_\alpha(x) \|_2 dx, \tag{7}$$

will be employed, as well as a curvature criterion

$$\kappa(x_0) := \frac{-1}{|B_0|} \int_{B_0} \nabla \cdot \frac{\nabla \log p_\alpha(x)}{\| \nabla \log p_\alpha(x) \|_2} dx. \tag{8}$$

Note the minus sign, which ensures that an inward pointing gradient (negative divergence) is associated with positive curvature, and vice versa. We choose $B_0$ to be the ball

$$B_0 = \{ x : \| \sqrt{1 - \alpha} x_0 - x \|_2 < \sqrt{d\alpha} \}. \tag{9}$$

This $B_0$ is practical: Its spherical boundary $\partial B_0$ (and its close neighbourhood) is highly probable under $p_\alpha$, and the score function on $\partial B_0$ can be approximated via the diffusion model at a fixed $t$ ( Equation (5))[3] [4]. Additionally, $\partial B_0$ is easily sampled as

$$\tilde{x} = \sqrt{1 - \alpha} x_0 + \sqrt{\alpha} u_d, \tag{10}$$

---

[3]This approximation is the *concentration of measure*, further reading: Giannopoulos & Milman (2000).
[4]To ensure $x_0 \in B_0$ we require $1 - \sqrt{1 - \alpha} < \sqrt{d\alpha} \|x_0\|_2$.

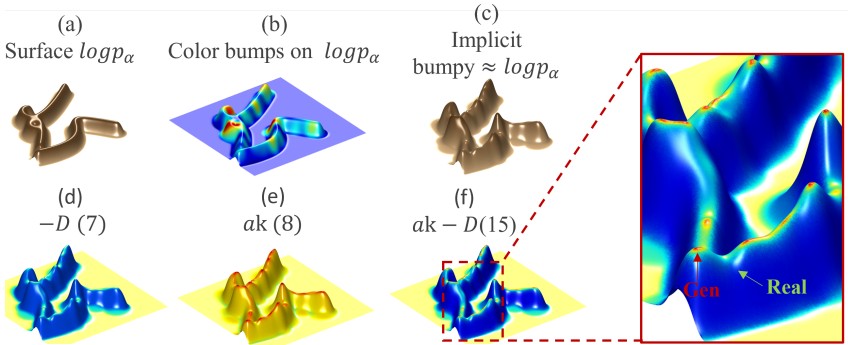

Figure 2: **Toy probability surface.** Simulation of toy data probability in a two-dimensional space ($d = 2$), structured along a one-dimensional manifold ($\Omega$ is a curve). (a) The log probability surface of perturbed samples, considering a uniform probability on the $\Omega$ curve. (b) A simulation of the hypothesis that generative models learn a bumpy version of the manifold: Bumps are randomly assigned to the manifold and visualized in color on the original surface. (c) The resulting bumpy surface. (d) Gradient magnitude of the bumpy manifold. (e) Total-variation curvature of the bumpy manifold. (f) Demonstrates the differential property derived from our analysis, highlighting locally maximal regions of the bumps which correspond to likely generated data points. We mathematically establish a way to capture this property through a zero-shot analysis of the diffusion model.

where $u_d \sim \text{Unif}(\mathcal{S}^{d-1}(\sqrt{d}))$ - a uniform distribution on the $(d-1)$-dimensional sphere centered at $\vec{0}$ with radius $\sqrt{d}$. Clarification: $\nabla \log p_{\alpha_t}(\cdot)$ can be applied to samples of $\tilde{x}$, yet the score function will be calculated for a sampled $x_t$, namely Equation (4), Equation (5) still hold, i.e.

$$\nabla \log p_\alpha(\tilde{x}) = \frac{1}{\alpha} \left( \sqrt{1-\alpha} \mathbf{E}_x[x_0|\tilde{x}] - \tilde{x} \right) \approx \frac{1}{\alpha} \left( \sqrt{1-\alpha} f(\tilde{x}, t; \theta) - \tilde{x} \right) := h(\tilde{x}), \quad (11)$$

where $h(\tilde{x})$ denotes the diffusion-model approximation of the score function. Notice the relation between $\tilde{x}$, constructed with $\alpha \leftarrow \alpha_t$ and $x_t$: As $d$ increases, the probability of $\|\epsilon\|_2$ ($\epsilon$ of Equation (3)) is concentrated around its mean $\sqrt{d}$, reducing the norm's stochasticity - making $u_d$ and $\epsilon$ (and as a consequence $\tilde{x}$ and $x_t$) interchangeable in high dimension $d$ Laurent & Massart (2000), see Fig. 4. [5]

### 4.2 MATHEMATICAL CLAIMS: ACCESSING KEY QUANTITIES AND CRITERIONS

**Claim 1.** *Given an image $x_0$ and a sample $x \sim \tilde{x}|x_0$, drawn according to Equation (10), we denote $u_d(x) = \frac{x - \sqrt{1-\alpha}x_0}{\sqrt{\alpha}}$. [6] Then the following relation holds:*

$$-\mathbf{E}_{x \sim \tilde{x}|x_0} \langle \frac{\nabla \log p_\alpha(x)}{\|\nabla \log p_\alpha(x)\|_2}, u_d(x) + \nabla \log p_\alpha(x) \rangle = \kappa(x_0) - D(x_0). \quad (12)$$

*This provides a characterization of $x_0$ as a stable maximal point under the backward diffusion process, Equation (6), quantifying both gradient magnitude (should be low) and curvature (should be high).*

*Proof Outline (full proof in the Appendix D)*

First we use Gauss divergence theorem for the curvature term

$$-|B_0|\kappa(x_0) = \int_{B_0} \nabla \cdot \frac{\nabla \log p_\alpha(x)}{\|\nabla \log p_\alpha(x)\|_2} dx = \int_{\partial B_0} \langle \frac{\nabla \log p_\alpha(x)}{\|\nabla \log p_\alpha(x)\|_2}, \hat{n} \rangle dx, \quad (13)$$

---

[5]Thus, model trained to denoise $x_t$, will perform well for $\nabla \log p_{\alpha_t}(\tilde{x}_t)$ in the sense of Equation (5), since $\tilde{x}_t$ samples from the highest-probability sub-sphere of $p(x_t)$. Infact, in our high-dimensional setting, sampling $x_t$ that is far away from $\partial B_0$ is highly improbable.

[6]The notation of $u_d$ as a function of $x$, i.e. $u_d(x)$, expresses the fact that $u_d(x)$, $x$ are constructed as a pair drawing the same noise. We use this when explicitly formulating expectation as an integral, where it is crucial to decide on one variable ($u_d$ or $x$) to integrate upon

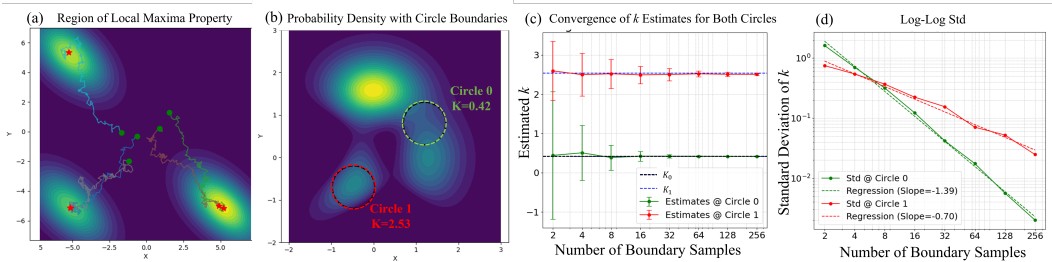

Figure 3: **a) The Local Maxima Region Property.** We trained a diffusion model on a 3-modal Gaussian Mixture Model (GMM) (details in Appendix B, implementation based on Brokman et al. (2025)). The colormap shows the learned PDF, with reverse diffusion trajectories overlaid. Starting points (green circles) converge toward local maxima of the probability(red stars). For statistics at scale see Fig. 7 **b) Expected Behavior of the Curvature Criterion $\kappa$.** We compute $\kappa$ on two marked circles centered at local maxima and saddle points of a differentiable analytic function (details in Appendix A). As expected, $\kappa$ is higher for the local maxima. **c) Error Analysis of $\kappa$ Estimators**. We experiment with both $\kappa$ values from b), and approximate them with increasing no. of spherical samples based on (Equation (13)). We average 100 runs and show std as error bars. Results confirm reliability: 1) The mean remains close to the true value even with few samples (unbiased estimator). 2) Separation between maxima and saddle points is maintained, even with as few as 4 samples. **d) Consistency and Convergence of $\kappa$ Estimators.** The standard deviation of $\kappa$ estimators is plotted against the number of spherical boundary samples in a log-log plot. Linear regression is applied to quantify the rate of convergence, showing a good fit with negative regression slopes, confirming exponential convergence. Combined with the empirical unbiasedness demonstrated in b), this establishes that the $\kappa$ estimators are empirically consistent.

where $\hat{n}$ denote the outward-pointing normals to the sphere $\partial B_0$. Using the properties of $\tilde{x}_t|x_0$, which is uniformly distributed on $\partial B_0$, we have that $\mathbf{E}_{x\sim\tilde{x}|x_0}(\cdot) = \frac{1}{|\partial B_0|}\int_{\partial B_0}(\cdot)$. Moreover, by construction $\hat{n} = \frac{x-\sqrt{1-\alpha}x_0}{\|\sqrt{1-\alpha}x_0-x\|_2} = \frac{u_d(x)}{\sqrt{d}}, \forall x \in \partial B_0$. Hence we get

$$\kappa(x_0) = -\frac{|\partial B_0|}{\sqrt{d}|B_0|}\mathbf{E}_{x\sim\tilde{x}_t|x_0}\big\langle\frac{\nabla\log p_\alpha(x)}{\|\nabla\log p_\alpha(x)\|_2}, u_d(x)\big\rangle. \tag{14}$$

We then use the same properties of the uniform distribution $\tilde{x}_t|x_0$, and obtain $D(x_0) = \mathbf{E}_{x\sim\tilde{x}_t|x_0}\big\langle\frac{\nabla\log p_\alpha(x)}{\|\nabla\log p_\alpha(x)\|_2}, \nabla\log p_\alpha(x)\big\rangle$. Finally - by properties of the hyper-sphere we have $\frac{|\partial B_0|}{\sqrt{d}|B_0|} = 1$, thus

$$-\mathbf{E}_{x\sim\tilde{x}|x_0}\big\langle\frac{\nabla\log p_\alpha(x)}{\|\nabla\log p_\alpha(x)\|_2}, u_d(x) + \nabla\log p_\alpha(x)\big\rangle = \kappa(x_0) - D(x_0). \tag{15}$$

**Corollary 2.** *In the setting of claim 1, we furthermore have the following approximation*

$$-\frac{\sqrt{1-\alpha}}{\alpha}\mathbf{E}_{x\sim\tilde{x}_t|x_0}\big\langle\frac{\nabla\log p_\alpha(x)}{\|\nabla\log p_\alpha(x)\|_2}, \hat{x}_0\big\rangle \approx \frac{1}{\sqrt{\alpha}}\kappa(x_0) - D(x_0). \tag{16}$$

*Proof Outline (full proof in the Appendix D)*

By linearity we decompose the expectation to summands via Equation (11). We have

$$\mathbf{E}_{x\sim\tilde{x}|x_0}\left(\frac{\nabla\log p_\alpha(x)}{\|\nabla\log p_\alpha(x)\|_2}\right) \approx 0, \tag{17}$$

since integration of normals over the sphere is zero, and $\nabla\log p_\alpha(x)$ approximates the uniform spherical noise. Thus, $\mathbf{E}_{x\sim\tilde{x}|x_0}\big\langle\frac{\nabla\log p_\alpha(x)}{\|\nabla\log p_\alpha(x)\|_2}, \sqrt{1-\alpha}x_0\big\rangle \approx 0$, since $x_0$ is deterministic. From here, dividing the (remaining two) summands by $\alpha$ leads to Equation 16, and we are done.

**Corollary 3.** *Let $b_0$ represent the statistical bias of predictor $\hat{x}_0$, defined as $b_0 = x_0 - \mathbf{E}_{x\sim\tilde{x}|x_0}(\hat{x}_0)$. Transitioning from score-function to the denoising perspective of diffusion models, the summand*

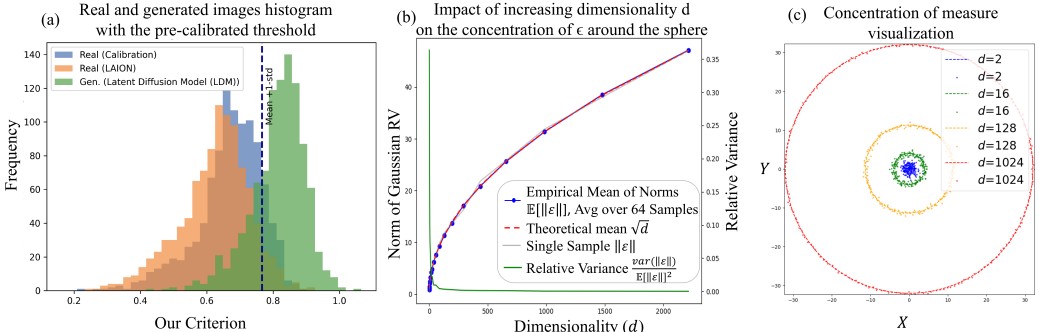

Figure 4: (a) We calibrate a decision threshold based on the mean and standard deviation of 1,000 real image criteria, ensuring it's free from generated data influence. Criteria from another dataset's real and generated images are also displayed. (b) As a result of $\|\epsilon\| \sim \chi$-distribution, as $d$ increases, $\epsilon$ concentrates around a spherical *thin shell*, with radii $\sqrt{d}$. This demonstrates interchangeable use of $x_t$ and $\tilde{x}$ in high-dimensional space. (c) 2D Visualization of the Concentration of Measure: For each $d$, the radii and samples' are set as the $d$-dimensional $\mathbb{E}\|\epsilon\|$ and $var(\|\epsilon\|)$ respectively. Correspondingly, the radii increases while the variance converges with $d$, effectively simulating the phenomenon in 2D.

*which is approximately zero in Corollary 2, given by $\mathbf{E}_{x \sim \tilde{x}|x_0} \left\langle \frac{\nabla \log p_\alpha(x)}{\|\nabla \log p_\alpha(x)\|_2}, x_0 \right\rangle$, satisfies*

$$-\frac{\alpha\sqrt{d}}{\sqrt{1-\alpha}}\mathbf{E}_{x \sim \tilde{x}|x_0} \left\langle \frac{\nabla \log p_\alpha(x)}{\|\nabla \log p_\alpha(x)\|_2}, x_0 \right\rangle \approx \langle b_0, x_0 \rangle, \tag{18}$$

*see proof in the Appendix. This is yet another quantity, capturing bias towards generating $x_0$, where:*

1. *This summand equals zero for unbiased noise predictors.*

2. *If not set to zero, higher values occur when $b_0$ is (anti-)correlated with $x_0$, indicating a bias in the (negative) noise prediction towards the clean image. It is intuitive that the noise prediction steers the iteratively denoised generated image towards its inherent biases - expressed by this summand.*

3. *In score function analysis settings, the noise predictor is approximated by an MMSE denoiser, which is unbiased, indeed leading to this summand being zero.*

### 4.3 NUMERICAL FORMULATION AND BEST PRACTICES

Given $x_0$, we approximate a three-term criterion of $\kappa(x_0), D(x_0), \langle b_0, x_0 \rangle$ via the approximations expressed in Corrs. 2, 3. The implementation, illustrated in Fig. 1, involves the following steps:
**1) Sampling Perturbations:** We generate $s$ spherical perturbations $\{u_d^{(i)}\}_{i=1}^s$ to produce samples $\{\tilde{x}^{(i)}\}_{i=1}^s$ according to Eq. Equation (10). These perturbations simulate variations around $x_0$ **2) Noise predictions:** Feed each perturbed sample $\tilde{x}^{(i)}$ to the diffusion model of choice to obtain noise predictions $h(\tilde{x}^{(i)})$ of Eq. Equation (11). **3) Criterion:** Compute:

$$C(x_0) := \frac{1}{s}\left(\sum_{i=1}^s \left\langle \frac{-h(\tilde{x}^{(i)})}{\|h(\tilde{x}^{(i)})\|_2}, au_d^{(i)} - bh(\tilde{x}^{(i)}) + c\sqrt{d}x_0 \right\rangle\right) \approx \frac{c_1}{\sqrt{\alpha}}\kappa(x_0) - c_2 D(x_0) + c_3 \langle b_0, x_0 \rangle,$$

where $a, b, c$ are scalars that determine the constants $c_1, c_2, c_3$. The factor $\frac{\sqrt{1-\alpha}}{\alpha}$ is omitted, as it is common across terms and can be absorbed into the selection of of $a, b, c$.

**Practical choices.** We map $u_d, h, x_0$ to CLIP Radford et al. (2021) before calculating $C(x_0)$. With stable diffusion, we first decode from latent space to image space, then map to CLIP. We set $a = b = c = 1$, though tuning is possible. In CLIP space, cosine similarity is used as the correct way to multiply embeddings. Dynamic range is adjusted to approximate $[0, 1]$ by scaling with $a + b + c$ and adding 1 (a minus sign instead of adding 1 will work, however it will flip the criteria order). For threshold calibration, we recommend using real images only to avoid bias, see Fig. 4.

Table 1: Comparison of zero-shot detection methods across various metrics. We report average AUC, AP, and Accuracy. Additionally, we include a top-10 generative technique performance, where for each detection method the 10 best-performing cases are selected, and all detection methods are compared on them. Our method significantly surpasses existing methods.

| Model | AUC | AP | Accuracy | RIGID Top 10 Accuracy | AEROBLADE Top 10 Accuracy | Ours Top 10 Accuracy |
|---|---|---|---|---|---|---|
| RIGID | 0.439 | 0.519 | 0.555 | 0.666 | 0.569 | 0.482 |
| AEROBLADE | 0.444 | 0.492 | 0.464 | 0.492 | 0.565 | 0.438 |
| Ours | **0.835** | **0.832** | **0.741** | **0.678** | **0.739** | **0.839** |

**Important take-away:** $C(x_0)$ approximates manifold-bias criteria. Another surprising perspective is that it measures similarity between the predictions of noise and data. Nevertheless, this is a result of our mathematical derivations and is supported hereafter by thorough evidence.

## 5 EVALUATION

In this section, we empirically validate our hypothesis: Bias-driven quantities, such as stable points in the generative process can serve as robust criteria for detecting generated images.

### 5.1 EXPERIMENTAL SETTINGS

**Datasets**. Our method is evaluated on a combination from three benchmark datasets featuring diverse generative techniques: **CNNSpot** Wang et al. (2020) comprises real and generated images from 20 categories of the LSUN Yu et al. (2015) dataset, featuring images produced by over ten generative models, primarily GANs. The **Universal Fake Detect** Ojha et al. (2023) dataset extends CNNSpot with generated images from newer models, primarily diffusion models. The **GenImage** Zhu et al. (2023) dataset features images produced by commercial generative tools, including Midjourney. In total, our aggregated dataset consists of $100K$ of real images and additional $100K$ images produced from 20 different generation techniques Karras et al. (2017); Zhu et al. (2017); Karras et al. (2019); Dhariwal & Nichol (2021); Ramesh et al. (2021); Rombach et al. (2022); Midjourney (2024). For the complete list of generative models used in our evaluation, see Appendix F.1.

**Implementation Details**.We used Stable Diffusion 1.4 Rombach et al. (2022) as our diffusion model and LLaVA 1.5 Liu et al. (2023) for generating text captions required as input by this model. Criterion hyper-parameters were set as follows: 1) No. of spherical noises $s$ was set to 64; 2) Perturbation strength $\alpha\sqrt{d} = 1.28$, determining $B_0$ radii and 3) A small scalar $\delta = 10^{-8}$ was added to the criterion denominator to ensure it is strictly positive. Code and datasets are detailed in Appendix C.

### 5.2 EMPIRICAL CASES AND RESULTS

We compare our method against SOTA zero-shot methods. We also test it in a mixture-of-experts (MoE), combined with a leading few-shot method. Ablation and sensitivity analyses are provided.

**Zero-shot Comparison Across** 20 **Generation Techniques**. In this experiment, we benchmark our method against two leading zero-shot image detectors: AEROBLADE Ricker et al. (2024) and RIGID He et al. (2024) under zero-shot settings. AEROBLADE Ricker et al. (2024) uses the reconstruction error for a given image obtained from a pre-trained variational autoencoder as the criterion for the detection task. RIGID He et al. (2024) compares the image representation of the original image and its noise-perturbed counterpart in a pre-trained feature space and uses their similarity as the detection criterion. Our implementation strictly adheres to the specifications detailed in their publications, utilizing their publicly available code. All methods used the same calibration set of $1K$ real images for threshold calibration and were evaluated with the test set described in Sec. 5.1, covering 20 diverse generation techniques to assess generalizability. More details in Appendix F.2.

Table 1 compares zero-shot detection methods across key performance metrics, showing the average score among all generative techniques. Our method outperforms existing methods by a significant margin. Fig. 5 provides an in-depth analysis featuring bar plots that summarize outcomes across

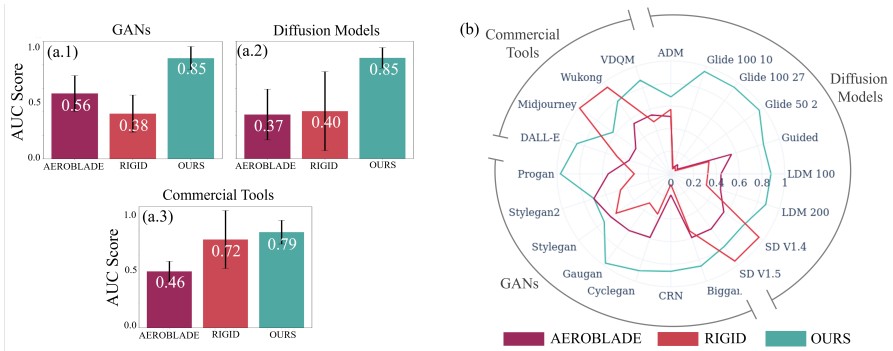

Figure 5: **Zero-shot comparison.** Plots $a.1$-$a.3$ demonstrate the superior AUC performance of our method across the three main generative technique groups. Error bars represent variability in AUC between techniques within each group, with our method showing the least variation. Plot $b$ details AUC per technique, where our method achieves the highest scores in most cases. Although our criterion originates from a zero-shot analysis of an LDM model, it empirically generalizes well to other techniques. Competitors show sensitivity to changes in technique, which hampers their generalization capabilities (clarified by detailed histograms in the Appendix, Fig. 8).

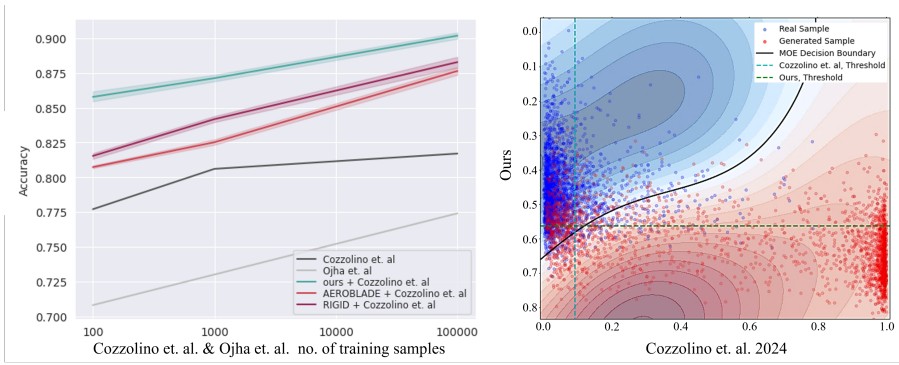

Figure 6: **Few-shot Performance**. Left - performance improvement in the MoE setting with Cozzolino et al. (2024a), in a few-shot regime. We report results of MoE with ours vs other zero-shot methods, and the original Ojha et al. (2023); Cozzolino et al. (2024a). Our efficacy proves to be the best. Right - scatter plot of our criterion and Cozzolino et al. (2024a). The decision boundary was obtained in our MoE setting using SVM. Significant improvement of separability can be observed.

various groups of generative techniques, including GANs, diffusion models, and commercial tools like Midjourney and DALL-E. This is complemented by a high-resolution polar plot comparing per-generative technique performance. Note: For the per-technique evaluation of Fig. 5 we have re-balanced the test sets per generative technique, making sure that each technique is evaluated on an equal number of real and generated images. Our method consistently outperforms AEROBLADE and RIGID across all groups and in the majority of generative techniques. Our competitors exhibit low AUC in generative techniques like Glide and CRN due to variations in their criteria across different generative techniques, see the per-technique histograms provided in the appendix, Fig. 8.

**Mixture of Experts with Few-shot Approaches.** While zero-shot scenarios often reflect real-world situations where data is unavailable or not worth managing, some cases may justify handling small amounts of generated data for significant performance gains. Recently, Cozzolino et al. (2024a) introduced a few-shot detection method leveraging a pre-trained CLIP. They established new benchmarks for generalization to unseen techniques in data-limited scenarios, reducing data maintenance costs yet not eliminating it altogether. To prove the applicability of zero-shot methods in the few-shot regime, we integrate them into a mixture-of-experts (MoE) framework alongside few-shot approaches, enhancing performance while remaining in the few-shot bounds. Utilizing an extra small set of examples, we trained a lightweight classifier to combine the outputs from a

Table 2: Sensitivity and ablation analyses. The table highlights the method's robustness across various configurations, including base models, number of perturbations $S$, spherical noise levels $\alpha$ (radii=$\alpha\sqrt{d}$), and image corruption techniques. The results show consistent performance with slight variations. Base settings are: SD v1.4 Model, $S = 64$, $\alpha = 0.01$ and without image corruption.

| Exp. | Number of Perturbations $S$ (SD v1.4 Model) | | | | | Level of Spherical Noises $\alpha$ (SD v1.4 Model) | | |
|---|---|---|---|---|---|---|---|---|
| Setting | 4 | 8 | 16 | 32 | 64 | $\alpha$=0.01 | $\alpha$=0.1 | $\alpha$=1 |
| AUC | 0.828 | 0.829 | 0.8308 | 0.833 | **0.835** | **0.835** | 0.82 | 0.815 |
| Exp. | Image Corruption Techniques (SD v1.4 Model) | | | | | Various Base Models | | |
| Setting | Without | Jpeg compression | | Gaussian blur | | Sd v1.4 | SD v2 base | Kandinsky 2.1 |
| AUC | **0.835** | 0.79 | | 0.822 | | **0.835** | 0.831 | 0.826 |

zero-shot method and Cozzolino et al. (2024a), forging a hybrid approach. In all MoE experiments, additional $1K$ labeled samples where used to train the light-weight classifier - these where randomly selected in an additional train-test split, implemented on the dataset initially used for zero-shot testing.

For the MoE approach we choose a random forest classifier. To visualize the MoE approach, we employ an SVM for its smoother decision boundary - see Fig. 6: While all zero-shot methods enhance performance, our method consistently outperforms others. For users willing to invest in managing a small amount of data, our method serves as an easy-to-integrate plugin that enhances few-shot frameworks, offering a trade-off between data availability and performance improvement.

**Sensitivity and Ablation Analysis**. We conducted evaluations to verify the robustness of our method across diverse configurations. This section provides key insights from these evaluations with the main results summarized in Table 2 and further details provided in the Appendix G.2. *Various stable diffusion models*. While our method focused on Stable Diffusion v1.4, here we also evaluated Stable Diffusion v2 Base and Kandinsky 2.1, which differ in size and technique. Results showed consistent performance with slight AUC decreases. *Various no. of perturbations $S$*. We explore different perturbation No. to assess their impact on the detection performance. The results reveal that increasing $S$ consistently enhances detection performance, which aligns with our research thesis. *Various spherical noise levels*. We varied the spherical noise levels (i.e., radii). These adjustments resulted in AUC decreases of 1.5% and 2%. *Image corruption techniques*. In real-world scenarios, adversaries may use compression or blurring to obscure traces of image generation. To address this, we evaluated our method under JPEG compression (AUC drop of 3.45%) and Gaussian blur (Kernel Size = 3, AUC drop of 1.2%), demonstrating its robustness to such techniques.

## 6 LIMITATIONS

Our derivations and the resulting quantities are induced by the learned manifold biases of *the analyzed diffusion model*. While detection of images generated by this model is expected, interestingly, our method shows good detection of generations from other models, some from completely different generative groups (Fig. 5). This cross-model capability is a notable strength, however there is no comprehensive theory to explain it. We hypothesize that different models, especially those trained on similar datasets, might exhibit similar characteristics within their probability manifolds and generated images Nalisnick et al. (2018); Kornblith et al. (2019). Further research is needed to substantiate this.

## 7 CONCLUSION

We introduced a novel zero-shot method to detect AI-generated images, capitalizing on biases inherent to the implicitly learned manifold of a pre-trained diffusion model. By combining score-function analysis with non-Euclidean manifold geometry, we deepen the theoretical understanding of manifold biases and use this framework to quantify differences between real and generated images.Our main hypothesis - that such bias-driven quantities can effectively detect generated content - has proven viable: Evaluations confirm that our method sets a new benchmark for zero-shot methods. Furthermore, we enhance few-shot performance - showing superior performance here as well. This establishes a foundation for further research into diffusion-model biases and their applications.

ACKNOWLEDGEMENTS

GG would like to acknowledge support by the Israel Science Foundation (Grant 1472/23) and by the Ministry of Science and Technology (Grant No. 5074/22).

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

## APPENDIX

In this Appendix, we provide additional information relevant to proofs, experimental settings, and experimental results. This document is presented as follows:

- **Reproducibility and Code** – links to our code project and dataset, as well as additional details regarding the implementation of our zero-shot method in our evaluation procedure.

- **Full Proofs** – detailed steps and justifications of our theoretical findings, which further support the claims presented in the main paper.

- **Experimental Settings Additional Information** – description regarding our used dataset and implementations.

- **Experimental Results Additional Information** – An extended analysis of the experimental results from the main manuscript.

## A  Experimental Details for $\kappa$ Error Analysis Experiments

In this appendix, we provide the full experimental details to re-produce Fig. 3(b-d) for estimating the curvature criterion $\kappa$ using spherical boundary samples. An extended version is available in Fig. 11 The experiments are designed to validate the approximation of $\kappa$ through averaging over the sphere, leveraging the Gauss Divergence Theorem. The results demonstrate the effectiveness of our method in distinguishing between local maxima and saddle points of a differentiable analytic function.

We define a two-dimensional differentiable analytic function $f : \mathbb{R}^2 \to \mathbb{R}$, inspired by MATLAB's `peaks` function, which exhibits multiple peaks and valleys. The function is given by:

$$f(x, y) = \frac{1}{C} \left[ 3(1 - x)^2 e^{-x^2 - (y+1)^2} - 10 \left( \frac{x}{5} - x^3 - y^5 \right) e^{-x^2 - y^2} - \frac{1}{3} e^{-(x+1)^2 - y^2} \right],$$

where $C$ is a normalization constant ensuring that the integral of $f$ over the domain is 1. Values of $f$ below a threshold (e.g., $1 \times 10^{-5}$) are set to zero to maintain non-negativity.

We select two circles centered at specific points to evaluate $\kappa$. The first circle is centered at a local maximum $(1.2, 0.8)$, and the second is centered at a saddle point $(-0.475, -0.7)$. Both circles have a radius of $R = 0.5$. These points represent distinct features of the function $f$, allowing us to assess the sensitivity of $\kappa$ to curvature differences.

For each circle, the true value of $\kappa$ is computed using the volume integral of Equation (8). Let $B_R(\mathbf{c})$ denote the circle of radius $R$ centered at $\mathbf{c}$. Numerical integration is performed by summing over the grid points inside each circle.

We approximate $\kappa$ using discrete samples along the boundary of each circle, discretizing Equation (13). The number of boundary samples $N_{\text{boundary}}$ is varied as $2, 4, 8, 16, 32, 64, 128, 256$. To reduce sampling bias, a random angular offset is introduced to the uniformly spaced boundary points in each run.

Boundary points are computed as:

$$(x_i, y_i) = (x_{\text{center}} + R \cos \theta_i, y_{\text{center}} + R \sin \theta_i),$$

where $\theta_i$ are the sampled angles. Gradients are interpolated at these boundary points, and the normalized gradient components are used to compute the dot product with the inward-pointing normal vector:

$$\mathbf{n}_{\text{in}} = -\frac{1}{R} \begin{pmatrix} x_i - x_{\text{center}} \\ y_i - y_{\text{center}} \end{pmatrix},$$

$$\left( \frac{\nabla f}{\|\nabla f\|} \cdot \mathbf{n}_{\text{in}} \right)_i = \left( \frac{\partial f}{\partial x} \right)_i^{\text{norm}} n_{x,i} + \left( \frac{\partial f}{\partial y} \right)_i^{\text{norm}} n_{y,i}.$$

The approximate $\kappa$ is then computed as:

$$\kappa_{\text{approx}} = \sum_{i=1}^{N_{\text{boundary}}} \left( \frac{\nabla f}{\|\nabla f\|} \cdot \mathbf{n}_{\text{in}} \right)_i \Delta s, \quad \Delta s = \frac{2\pi R}{N_{\text{boundary}}}.$$

For each $N_{\text{boundary}}$, 100 independent runs are performed to compute the mean and standard deviation of $\kappa_{\text{approx}}$. Error bars represent the standard deviation.

The results validate the approximation, showing that the mean of $\kappa_{\text{approx}}$ aligns with the true $\kappa$, demonstrating unbiasedness. The variance of $\kappa_{\text{approx}}$ decreases as $N_{\text{boundary}}$ increases, indicating improved accuracy with more samples. Even with low $N_{\text{boundary}}$, the estimated $\kappa$ values for the local maximum and saddle point are distinctly separated within error margins, supporting the reliability of the method.

## B  Experimental Detals for the Local Probability Maxima Property Verification.

In this section, we present a the experimental details of Fig. 3 (a) to illustrate how generated samples from a diffusion model tend to converge to stable local maxima on the learned probability manifold.

The learned manifold is approximated using Kernel Density Estimation (KDE) of the generated samples. Statistics at scale are provided below, Fig. 7.

We constructed a synthetic dataset by sampling from a Gaussian Mixture Model (GMM) in 2D with the following parameters: means $= \left\{ \begin{pmatrix} -5 \\ -5 \end{pmatrix}, \begin{pmatrix} 0 \\ -5 \end{pmatrix}, \begin{pmatrix} -5 \\ 0 \end{pmatrix} \right\}$., covariances $= \left\{ \begin{pmatrix} 0.1 & 0 \\ 0 & 0.1 \end{pmatrix}, \begin{pmatrix} 0.1 & 0 \\ 0 & 0.1 \end{pmatrix}, \begin{pmatrix} 0.1 & 0 \\ 0 & 0.1 \end{pmatrix} \right\}$. and weights $= \left\{ \frac{1}{3}, \frac{1}{3}, \frac{1}{3} \right\}$. For the training set we produce 1000 training data points from the defined GMM.

We trained a diffusion model on this dataset to learn the underlying data distribution. The key components of the training process we used: $T = 100$ diffusion steps and define a linear noise schedule with $\beta_t$ linearly spaced between $1 \times 10^{-4}$ and 0.02. The denoising model is a simple neural network consisting of fully connected layers with ReLU activation. Standard diffusion model training is done for $n_{\text{epochs}} = 1000$ epochs, were a a simple MSE loss between predicted and true noise is used.

We use the trained diffusion model to generate $n_{\text{gen\_samples}} = 1000$ new samples. For a random subset of $n_{\text{trajectories}} = 5$ samples, we record their trajectories during the reverse diffusion process to analyze their paths towards convergence. Furthermore, to approximate the learned probability manifold, we apply Kernel Density Estimation (KDE) on the generated samples. In Fig. 4 (a) it is indeed observed - that the generation trajectories terminate at local maximas

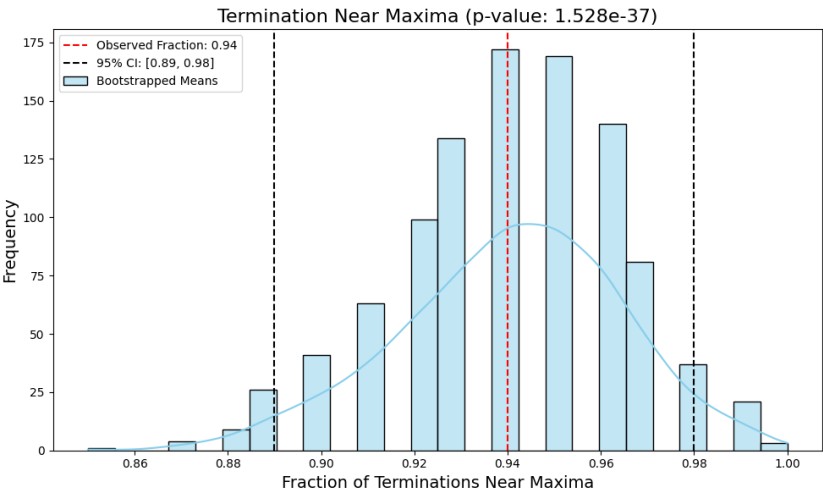

Figure 7: **Termination Analysis of Diffusion Trajectories Near Local Maxima.** The plot shows the fraction of 100 diffusion trajectories terminating near one of the maxima of the Gaussian mixture model (GMM). A trajectory is considered to terminate near a local maximum if its final point lies within a Mahalanobis distance of 2.45 from any of the GMM component means, corresponding to approximately 95% of the mass of a 2D Gaussian. The red dashed line represents the observed termination-near-maxima fraction (0.94), while the black dashed lines indicate the 95% confidence interval (0.89 to 0.98), derived via bootstrapping. In the bootstrapping process, the termination data (binary values indicating whether each of the 100 trajectories terminates near a maximum) was resampled with replacement 1,000 times to compute the distribution of termination fractions. The histogram and KDE curve illustrate this bootstrapped distribution. The p-value from a binomial test ($p = 1.528 \times 10^{-37}$) confirms that the observed termination fraction significantly deviates from random chance, supporting strong convergence of trajectories toward maxima.

## C   REPRODUCIBILITY AND CODE

To ensure reproducibility, we provide our code and a detailed description of our computational environment.

**Code.** The implementation of our zero-shot detection method, as well as the train and test sets are available at the following link: `https://tinyurl.com/zeroshotimplementation`. To reproduce our results follow the readme file. **Hardware and Programs.** All of the experiments were conducted on the Ubuntu 20.04 Linux operating system, equipped with a Standard NC48ads A100 v4 configuration, featuring 4 virtual GPUs and 440 GB of memory. The experimental code base was developed in Python 3.8.2, utilizing PyTorch 2.1.2 and the NumPy 1.26.3 package for computational tasks.

## D   MATHEMATICAL FULL PROOFS

**Claim 1.** *Given $x_0$, consider Equation (10), for which samples $x \sim \tilde{x}|x_0$ are drawn uniformly from the sphere $\partial B_0$, and each $x$ is drawn with a corresponding $u_d(x) = \frac{x - \sqrt{1-\alpha}x_0}{\sqrt{\alpha}}$.[7] Then we can use $\nabla log p_\alpha(\cdot)$ to obtain*

$$-\mathbf{E}_{x \sim \tilde{x}|x_0} \langle \frac{\nabla \log p_\alpha(x)}{\|\nabla \log p_\alpha(x)\|_2}, u_d(x) + \nabla \log p_\alpha(x) \rangle = \kappa(x_0) - D(x_0). \tag{19}$$

*This provides a characterization of $x_0$ as a stable maximal point under the backward diffusion process, Equation (6), quantifying both gradient magnitude (should be low) and curvature (should be high) aspects.*

*Proof.* Let us begin with the curvature term. By Gauss divergence Thm. we have

$$-|B_0|\kappa(x_0) = \int_{B_0} \nabla \cdot \frac{\nabla \log p_\alpha(x)}{\|\nabla \log p_\alpha(x)\|_2} dx = \int_{\partial B_0} \langle \frac{\nabla \log p_\alpha(x)}{\|\nabla \log p_\alpha(x)\|_2}, \hat{n} \rangle dx, \tag{20}$$

where $\hat{n}$ are the outward-pointing normals to the sphere $\partial B_0$. By construction we have $\hat{n} = \frac{\sqrt{1-\alpha}x_0 - x}{\|\sqrt{1-\alpha}x_0 - x\|_2} = \frac{u_d(x)}{\sqrt{d}}$, $\forall x \in \partial B_0$, thus

$$\int_{\partial B_0} \langle \frac{\nabla \log p_\alpha(x)}{\|\nabla \log p_\alpha(x)\|_2}, \hat{n} \rangle dx = \frac{1}{\sqrt{d}} \int_{\partial B_0} \langle \frac{\nabla \log p_\alpha(x)}{\|\nabla \log p_\alpha(x)\|_2}, u_d(x) \rangle dx. \tag{21}$$

Using the properties of $\tilde{x}_t|x_0$, which is uniformly distributed on $\partial B_0$, we have

$$\frac{1}{\sqrt{d}} \int_{\partial B_0} \langle \frac{\nabla \log p_\alpha(x)}{\|\nabla \log p_\alpha(x)\|_2}, u_d(x) \rangle dx = \frac{|\partial B_0|}{\sqrt{d}} \mathbf{E}_{x \sim \tilde{x}_t|x_0} \langle \frac{\nabla \log p_\alpha(x)}{\|\nabla \log p_\alpha(x)\|_2}, u_d(x) \rangle, \tag{22}$$

where $|\partial B_0|$ denotes the volume of $\partial B_0$. Tracing this back to $\kappa$ (Equation (20)) we have

$$\kappa(x_0) = -\frac{|\partial B_0|}{\sqrt{d}|B_0|} \mathbf{E}_{x \sim \tilde{x}_t|x_0} \langle \frac{\nabla \log p_\alpha(x)}{\|\nabla \log p_\alpha(x)\|_2}, u_d(x) \rangle. \tag{23}$$

Finally, we use the known formula for the ratio between a $d$-dimensional ball's volume and its boundary-sphere volume - plugging radii $\sqrt{d}$ as follows

$$\frac{\partial B_0}{|B_0|} = \frac{d}{\text{radii}} = \sqrt{d},$$

resulting with $\frac{|\partial B_0|}{\sqrt{d}|B_0|} = 1$. Thus

$$\kappa(x_0) = -\frac{|\partial B_0|}{\sqrt{d}|B_0|} \mathbf{E}_{x \sim \tilde{x}_t|x_0} \langle \frac{\nabla \log p_\alpha(x)}{\|\nabla \log p_\alpha(x)\|_2}, u_d(x) \rangle. \tag{24}$$

Let us now analyze the gradient term. We similarly use the properties of the uniform distribution $\tilde{x}_t|x_0$, and get

$$D(x_0) = \frac{1}{|\partial B_0|} \int_{\partial B_0} \|\nabla \log p_\alpha(x)\|_2 dx = \mathbf{E}_{x \sim \tilde{x}_t|x_0} \|\nabla \log p_\alpha(x)\|_2. \tag{25}$$

---

[7]The notation of $u_d$ as a function of $x$, i.e. $u_d(x)$, expresses the fact that they are constructed using the same noise. We use this under the integration sign when expressing explicitly expectations, where it is crucial to decide on one variable to integrate upon

For convenience, let us plug $\|\nabla \log p_\alpha(x)\|_2 = \langle \frac{\nabla \log p_\alpha(x)}{\|\nabla \log p_\alpha(x)\|_2}, \nabla \log p_\alpha(x) \rangle$ and get

$$D(x_0) = \mathbf{E}_{x \sim \tilde{x}_t | x_0} \langle \frac{\nabla \log p_\alpha(x)}{\|\nabla \log p_\alpha(x)\|_2}, \nabla \log p_\alpha(x) \rangle. \tag{26}$$

Finally, by linearity

$$-\mathbf{E}_{x \sim \tilde{x} | x_0} \langle \frac{\nabla \log p_\alpha(x)}{\|\nabla \log p_\alpha(x)\|_2}, u_d(x) + \nabla \log p_\alpha(x) \rangle = \kappa(x_0) - D(x_0). \tag{27}$$

$\square$

**Corollary 2.** *In the setting of claim 1, we furthermore have the following approximation*

$$-\mathbf{E}_{x \sim \tilde{x}_t | x_0} \langle \frac{\nabla \log p_\alpha(x)}{\|\nabla \log p_\alpha(x)\|_2}, \frac{\sqrt{1-\alpha}}{\alpha} \hat{x}_0 \rangle \approx \frac{1}{\sqrt{\alpha}} \kappa(x_0) - D(x_0). \tag{28}$$

*Proof.* By the score-function approximation as in Equation (11), and due to $\epsilon$ and $u_d$ being approximately interchangeable in our high-dimensional setting, we have $\sqrt{1-\alpha}\hat{x}_0 = \tilde{x} + h(\tilde{x}) \approx \sqrt{1-\alpha}x_0 + \sqrt{\alpha}u_d + \alpha log p_\alpha(\tilde{x})$. Thus

$$\sqrt{1-\alpha}\mathbf{E}_{x \sim \tilde{x} | x_0} \langle \frac{\nabla \log p_\alpha(x)}{\|\nabla \log p_\alpha(x)\|_2}, \hat{x}_0 \rangle \approx \mathbf{E}_{x \sim \tilde{x} | x_0} \langle \frac{\nabla \log p_\alpha(x)}{\|\nabla \log p_\alpha(x)\|_2}, \sqrt{1-\alpha}x_0 + \sqrt{\alpha}u_d(x) + \alpha \nabla \log p_\alpha(x) \rangle \tag{29}$$

Notice that

$$\mathbf{E}_{x \sim \tilde{x} | x_0} \langle \frac{\nabla \log p_\alpha(x)}{\|\nabla \log p_\alpha(x)\|_2}, \sqrt{1-\alpha}x_0 \rangle = \langle \mathbf{E}_{x \sim \tilde{x} | x_0} \left( \frac{\nabla \log p_\alpha(x)}{\|\nabla \log p_\alpha(x)\|_2} \right), \sqrt{1-\alpha}x_0 \rangle \approx 0, \tag{30}$$

since integration of normals over the sphere is zero, and $\nabla \log p_\alpha(x)$ approximates the uniform spherical noise. Thus by linearity of the expectation

$$\mathbf{E}_{x \sim \tilde{x} | x_0} \langle \frac{\nabla \log p_\alpha(x)}{\|\nabla \log p_\alpha(x)\|_2}, \sqrt{1-\alpha}x_0 + \sqrt{\alpha}u_d(x) + \alpha \nabla \log p_\alpha(x) \rangle = \mathbf{E}_{x \sim \tilde{x} | x_0} \langle \frac{\nabla \log p_\alpha(x)}{\|\nabla \log p_\alpha(x)\|_2}, \sqrt{\alpha}u_d(x) + \alpha \nabla \log p_\alpha(x) \rangle. \tag{31}$$

Tracing back to Equation (29) and dividing by $\alpha$, we get

$$\frac{\sqrt{1-\alpha}}{\alpha} \mathbf{E}_{x \sim \tilde{x}_t | x_0} \langle \frac{\nabla \log p_\alpha(x)}{\|\nabla \log p_\alpha(x)\|_2}, \hat{x}_0 \rangle \approx \mathbf{E}_{x \sim \tilde{x} | x_0} \langle \frac{\nabla \log p_\alpha(x)}{\|\nabla \log p_\alpha(x)\|_2}, \frac{1}{\sqrt{\alpha}} u_d(x) + \nabla \log p_\alpha(x) \rangle. \tag{32}$$

Finally, similarly to Equation (27), and by linearity of the expectation, we get

$$-\frac{\sqrt{1-\alpha}}{\alpha} \mathbf{E}_{x \sim \tilde{x}_t | x_0} \langle \frac{\nabla \log p_\alpha(x)}{\|\nabla \log p_\alpha(x)\|_2}, \hat{x}_0 \rangle \approx \frac{1}{\sqrt{\alpha}} \kappa(x_0) - D(x_0). \tag{33}$$

$\square$

**Corollary 3.** *Let $b_0$ represent the bias of predictor $\hat{x}_0$ in the statistical sense, defined as $b_0 = x_0 - \mathbf{E}_{x \sim \tilde{x} | x_0}(\hat{x}_0)$. Transitioning from score-function back to the denoising perspective of diffusion models, the zeroized summand of Corollary 2, given by $\mathbf{E}_{x \sim \tilde{x} | x_0} \left\langle \frac{\nabla \log p_\alpha(x)}{\|\nabla \log p_\alpha(x)\|_2}, x_0 \right\rangle$, can be traced back to:*

$$\mathbf{E}_{x \sim \tilde{x} | x_0} \left\langle \frac{\nabla \log p_\alpha(x)}{\|\nabla \log p_\alpha(x)\|_2}, x_0 \right\rangle \approx -\frac{1}{\alpha\sqrt{d}} \langle b_0, \sqrt{1-\alpha}x_0 \rangle, \tag{34}$$

*which is yet another quantity that captures bias towards generating $x_0$, where:*

1. *This summand is effectively zeroized for unbiased noise predictors.*

2. *If not zeroized, higher values occur when $b$ is (anti-)correlated with $x_0$, indicating a bias in the (minus) noise prediction towards the clean image. Since diffusion models denoise pure noise - it is intuitive that the noise prediction steers the resulting generated image towards its inherent biases - as is captured by this summand.*

3. *In score function analysis settings, the noise predictor is approximated by an MMSE denoiser, which is assumed to be unbiased, indeed leading to the zeroizing of this summand.*

*Proof.*

$$-\mathbf{E}_{x\sim\tilde{x}|x_0}\left\langle\frac{\nabla\log p_\alpha(x)}{\|\nabla\log p_\alpha(x)\|_2},x_0\right\rangle \approx \mathbf{E}_{x\sim\tilde{x}|x_0}\left(\frac{\frac{1}{\alpha}\langle x-\sqrt{1-\alpha}\hat{x}_0,x_0\rangle}{\frac{1}{\alpha}\|x-\sqrt{1-\alpha}\hat{x}_0\|}\right)$$

$$\approx \mathbf{E}_{x\sim\tilde{x}|x_0}\left(\frac{\frac{1}{\alpha}\langle x-\sqrt{1-\alpha}\hat{x}_0,x_0\rangle}{\frac{1}{\alpha}\sqrt{\alpha d}}\right)$$

$$= \frac{1}{\sqrt{d}}\langle\mathbf{E}_{x\sim\tilde{x}|x_0}(\sqrt{1-\alpha}x_0+u_d-\sqrt{1-\alpha}\hat{x}_0),x_0\rangle$$

$$= \frac{1}{\alpha\sqrt{d}}\langle\sqrt{1-\alpha}x_0+\mathbf{E}_{x\sim\tilde{x}|x_0}(u_d)-\sqrt{1-\alpha}\mathbf{E}_{x\sim\tilde{x}|x_0}(\hat{x}_0),x_0\rangle$$

$$= \frac{\sqrt{1-\alpha}}{\alpha\sqrt{d}}\langle x_0+0-\mathbf{E}_{x\sim\tilde{x}|x_0}(\hat{x}_0),x_0\rangle$$

$$= \frac{\sqrt{1-\alpha}}{\alpha\sqrt{d}}\langle x_0-\mathbf{E}_{x\sim\tilde{x}|x_0}(\hat{x}_0),x_0\rangle$$

$$= \frac{\sqrt{1-\alpha}}{\alpha\sqrt{d}}\langle b_0,x_0\rangle$$

First transition follows Equation (4), Equation (5) and plugs Equation (10). The second transition uses the noise estimation perspective, where $\alpha u_d$ perturbation predictions should have approximately $\alpha\sqrt{d}$ norm.

The other transitions use linearity of the expectation, and the zero mean of the spherical noise $u_d$.

$\square$

# E   APPROXIMATION DISCUSSION

Reminder:

$$D(x_0) = \frac{1}{|\partial B_0|}\int_{\partial B_0}\|\nabla\log p_\alpha(x)\|_2 dx = \mathbf{E}_{x\sim\tilde{x}_t|x_0}\|\nabla\log p_\alpha(x)\|_2. \tag{35}$$

$$\kappa(x_0) = -\frac{|\partial B_0|}{\sqrt{d}|B_0|}\mathbf{E}_{x\sim\tilde{x}_t|x_0}\left\langle\frac{\nabla\log p_\alpha(x)}{\|\nabla\log p_\alpha(x)\|_2},u_d(x)\right\rangle. \tag{36}$$

$$\nabla\log p_\alpha(\tilde{x}) = \frac{1}{\alpha}\left(\sqrt{1-\alpha}\mathbf{E}_x[x_0|\tilde{x}]-\tilde{x}\right) \approx \frac{1}{\alpha}\left(\sqrt{1-\alpha}f(\tilde{x},t;\theta)-\tilde{x}\right) := h(\tilde{x}), \tag{37}$$

$$C(x_0) := \frac{\sqrt{1-\alpha}}{\alpha}\frac{1}{s}\sum_{i=1}^{s}\left\langle\frac{-h(\tilde{x}^{(i)})}{\|h(\tilde{x}^{(i)})\|_2},\hat{x}_0\right\rangle \approx a\kappa(x_0)-D(x_0),$$

# F   EXPERIMENTAL SETTINGS ADDITIONAL INFORMATION

## F.1   DATASETS

As mention in the main manuscript, the evaluation of our proposed method incorporates three benchmark datasets, namely, CNNSpot Wang et al. (2020), Universal Fake Detect Ojha et al. (2023) and GenImage Zhu et al. (2023) datasets. In our evaluation, we extracted a subset from each dataset, containing real images and fake images generated from the following generative models: ProGAN Karras et al. (2017), StyleGAN Karras et al. (2019), BigGAN Brock et al. (2018),GauGAN Park et al.

(2019), CycleGAN Zhu et al. (2017), StarGAN Choi et al. (2018), Cascaded Refinement Networks (CRN) Chen & Koltun (2017), Implicit Maximum Likelihood Estimation (IMLE) Li et al. (2019), SAN Dai et al. (2019), seeing-dark Chen et al. (2018), deepfake Rossler et al. (2019), Midjourney Midjourney (2024), Stable Diffusion V1.4 Rombach et al. (2022), Stable Diffusion V1.5 Rombach et al. (2022), ADM Dhariwal & Nichol (2021), Wukong MindSpore (2024), VQDM Gu et al. (2022), LDM Rombach et al. (2022) and Glide Nichol et al. (2021). In Figure 5 in the main manuscript, we divided our dataset into three groups: images generated by GANs (produces by ProGAN, StyleGAN, StyleGan2, BigGAN, GauGAN, CycleGAN, and CRN models), diffusion models(produces by LDM, Glide, Stable Diffusion V1.4, Stable Diffusion V1.5 and Guided diffusion models), and commercial tools (produces by Midjourney, Wukong, VQDM and DALL-E tools). Additionally, for the real images we used the LSUN, MSCOCO, ImageNet and LAION datasets.

To construct the calibration set for the zero-shot methods, we extracted 1,000 real samples from the datasets. For the test set, we selected 200,000 samples, ensuring a representative volume from each generation technique.

### F.2    Methods for Comparison

We benchmarked our method against two recent and leading image detection zero-shot methods Ricker et al. (2024); He et al. (2024). These state-of-the-art methods are designed to enhance generalization by detecting generated images in a zero-shot settings. The implementations closely follow the specifications outlined in their respective publications. Specifically, we applied Ricker et al. (2024) directly by employing the code provided in their published paper, selecting the parameters leading to the highest performance according to their report (such as using the Kandinsky 2.1 model with the LPIPS similarity metric). Since He et al. (2024) was not available at the time of paper submission, we carefully reconstructed their implementation based on the details they provided, applying identical parameters, such as applying the DINO model with perturbation noise level of $0.05$ and threshold value of $95\%$.

In the mixture-of-expert (MoE) experiment we utilized two additional leading image detection methods Cozzolino et al. (2024a); Ojha et al. (2023). These few-shot and semi-supervised methods are designed to enhance generalization in detecting images created by unseen generative techniques. The implementations closely follow the specifications outlined in their respective publications. Specifically, the detection models are trained on images generated by a single model (ProGAN Karras et al. (2017) from the CNNSpot Wang et al. (2020) dataset) and tested on images from various other models. In implementing both methods, we initially employed the CLIP embedder Radford et al. (2021) using the open-source "clip-vit-large-patch14" model. For Ojha et al. (2023), we utilized a KNN model with $k = 9$ and cosine similarity, as this configuration was reported to achieve the best results in their paper. For Cozzolino et al. (2024a), we employed a standard SVM model Pedregosa et al. (2011). Training of both methods was conducted with 10 different seeds $(1, 5, 9, 16, 17, 24, 43, 54, 59, 65)$, and the final detection results were averaged to ensure robustness.

## G    Experimental Results Additional Information

### G.1    Complementary Results

In Fig. 8 statistics are gathered from all zero-shot methods to shed light on the variability of the different criteria (ours, RIGID, and AEROBLADE) across generative techniques. The competitors exhibit high variability, demonstrating their lack of generalizability. In the main text, under 5 in some cases the competitors have surprisingly low AUCs . This is due to their criteria being overly sensitive - as shown in the attached histograms. While in some techniques generated images obtain higher criteria values compared to real ones (e.g., Wukong), others (e.g., Glide 50) demonstrate a reverse trend. This makes the setting of a global threshold, such as the proposed $95\%$ percentile of real images in RIGID, ineffective in some generative techniques it became - reducing their overall AUC. While incorporating information about the specific generative technique could improve their performance, it would compromise real-world practicality, and violate the standard testing for generalization to unseen techniques.

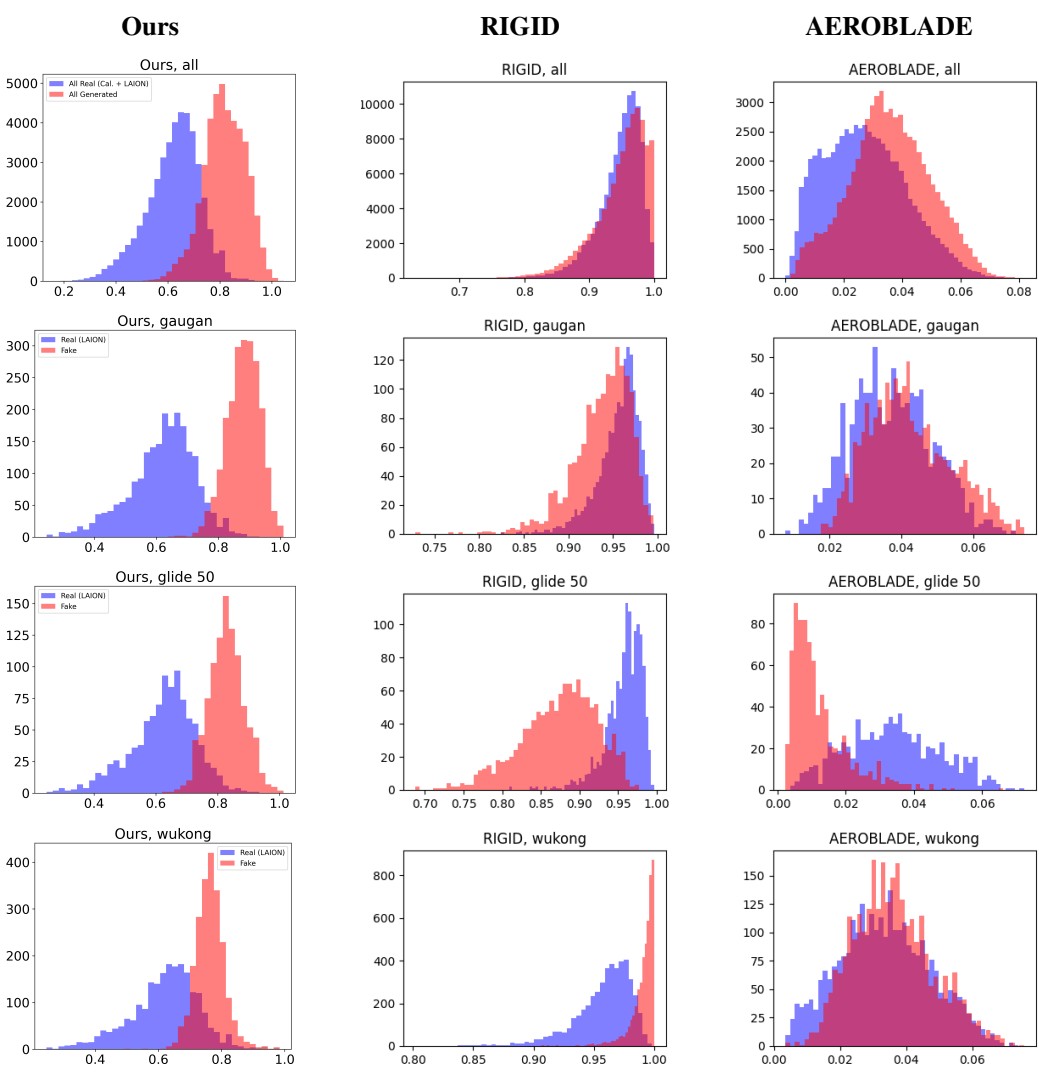

Figure 8: Histograms of the criterion proposed by our method and the competitors, for all the data as well as per-generative-technique cases. While in some cases the competitors exhibit high separability between real (blue) and generated (red, "fake") images, their inconsistent behavior—where the real histograms are sometimes below and sometimes above the fake —prevents setting a reliable global threshold. This inconsistency makes them less suitable for the task of generalization to unseen techniques, where we assume that the tested technique is unknown. For instance, RIGID's incosistency between Wukong and glide 50 observed here, translates in the polar-plot of Fig. 5(b) to good AUC for Wukong, but almost opposite (close to zero) AUC for Glide 50. In contrast, our method exhibits consistent behavior across generative techniques, which is a hallmark of good cross-technique generalization.

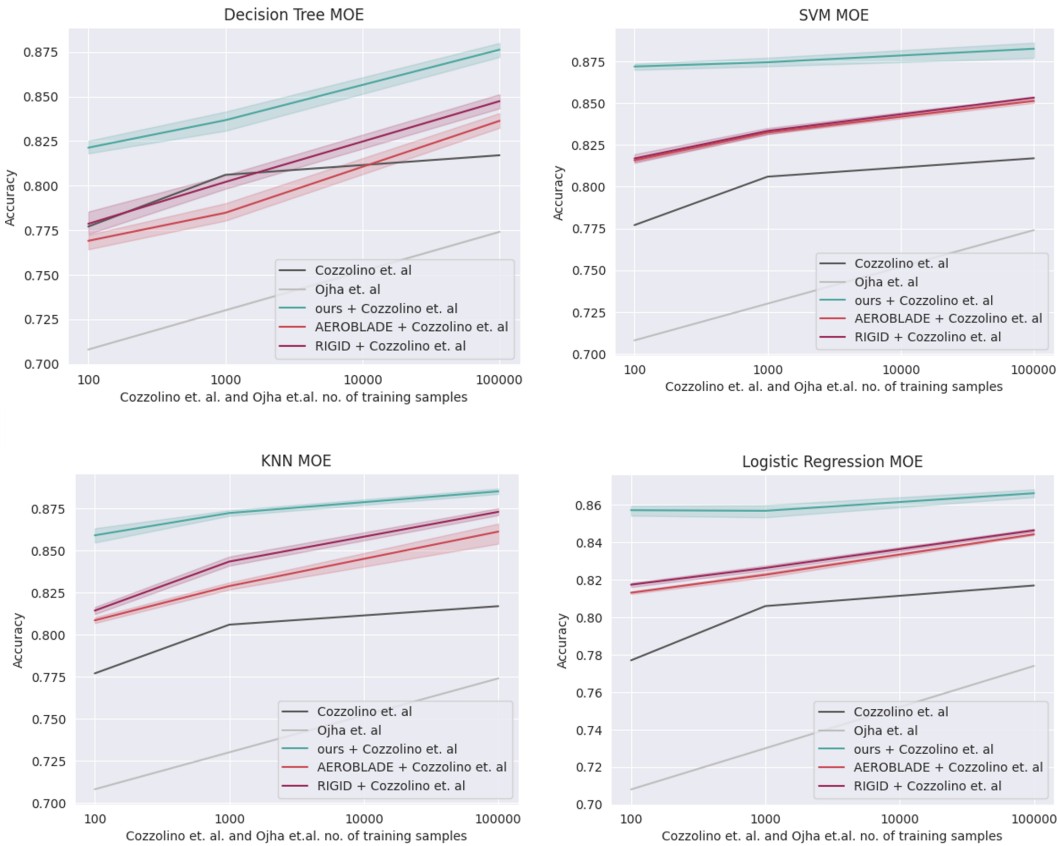

Figure 9: **Few-shot MoE**. Results of our MoE experiment using different classification algorithms. All combinations outperform Cozzolino significantly. Our method's efficacy proves to be the best.

## G.2 SENSITIVITY AND ABLATION ANALYSIS ADDITIONAL INFORMATION

**Various stable diffusion models.** Our approach exploits the implicitly learned probability manifold of diffusion models to distinguish AI-generated images. While we demonstrated the robustness of our methodology using the Stable Diffusion v1.4 model, it is critical to verify that our results are not unduly influenced by specific characteristics of the used model. Consequently, we broadened our assessment scope to include newer models such as Stable Diffusion v2 Base and Kandinsky 2.1, noting these versions exhibit variations in architecture scale and generative algorithms. The methodology deployed in these expanded evaluations was the same as our original experiments (described in Sec. 5.1 of our manuscript), maintaining consistency in the calibration and test sets employed. This consistent experimental framework ensures that any observed performance variations are attributable solely to the model differences rather than experimental conditions. Preliminary findings indicate a modest performance decrement of approximately 2% with both Stable Diffusion v2 Base and Kandinsky 2.1, suggesting slight variances in generative fidelity and stability across model versions.

**Various classification algorithms for MoE settings.** In the main manuscript, we present the results of our MoE experiment when using a random forest algorithm to combine the results of the methods used in the MoE. Fig. 9 presents the results when different classification algorithms are used: decision tree, support-vector-machine (SVM), K-nearest-neighbors (KNN), and logistic regression. All models were set using the default parameters of the scikit-learn Python library. As can be seen in Fig. 9, our method's efficacy proves to be the best regardless of the algorithm used.

**Various no. of perturbations** $S$. One of the hyperparameters of our method is the volume of perturbations. In this experiment, we explore different perturbation sizes to assess their impact on the detection performance. For consistency, we maintained the same experiment methodologies

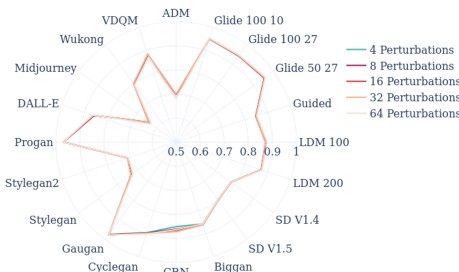 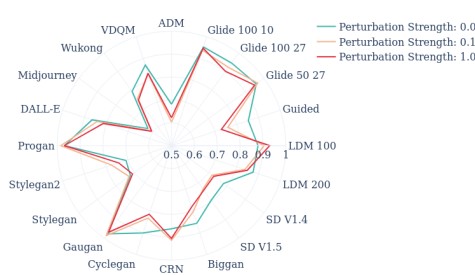

Figure 10: **AUC Using Varying Hyper-parameter Values, across Different Generation Techniques: The hyper-parameters sensitivity analysis, summarized in Table 2, detailed per-model**. Left: The number of perturbations ($s$) shows minimal impact on performance. Right: Varying perturbation strengths ($\alpha$). Interestingly, performance is affected differently across models.

as described in Sec. 5.1 of our manuscript, focusing solely on variations in perturbation size. We tested perturbation sizes of $4$, $8$, $16$, $32$, and $64$, as implemented in our method. The corresponding average AUC detection results were $0.828$, $0.829$, $0.830$, $0.833$, and $0.835$ respectively. The results reveal a clear trend: increasing $S$ consistently enhances detection performance, which aligns with our research thesis. These findings reveal a clear trend: as the number of perturbations $S$ increases, so does the detection performance. This progression supports our research theory since higher volume of perturbations provide better approximations of the numerical quantities.

*Various spherical noise levels*. Another hyperparameter in our method involves the levels of spherical noise, specifically the radii. In this experiment, we systematically adjusted the noise by reducing the radii, first by a factor of 10 and subsequently by a factor of 100. Additionally, we scaled down the test set to $16,000$ images for this small-scale study, though the variety of generative techniques assessed remained constant. These modifications led to performance decreases of 1.5% and 2%, respectively. These results help us understand the impact of noise adjustments on the robustness of our detection capabilities.

**Robustness to JPEG**. In real-world scenarios, images are frequently compressed to JPEG format for easier storage and transmission. To assess our method's practicality, we evaluated its performance on both real and generated JPEG-compressed images. For this small-scale study, we reduced the test set to 16,000 images, while maintaining the same variety of generative techniques. The results indicated a modest decrease in accuracy of 3.45%, demonstrating minimal impact on the method's detection effectiveness

## G.3 RUN-TIME ANALYSIS

We performed runtime analysis to evaluate the computational efficiency of our method compared to existing approaches. Using a single A100 GPU, we could process in parallel a batch of 4 samples using 16 perturbations, and observed a runtime of **2.1 seconds per sample**. Importantly, this implies a 64-perturbation setup can be fully parallelized on a single A100 GPU with batches of 1. For comparison, our primary competitor, AEROBLADE, on the same A100 GPU requires **5.4 seconds per sample**, making our method significantly more computationally efficient in terms of runtime. We do note AEROBLADE's official implementation at the time of the experiment was not compatible with batch parallel processing - thus we believe it could be sped up.

## G.4 DETAILED ZERO-SHOT COMPARISON

Tables 3 shows the entire zero-shot comparison for each detection method, over each detection technique using the accuracy, area-under-the-curve (AUC) and average precision (AP) metrics.

Table 3: Performance metrics across different models and methods

| Model | Accuracy | | | AUC | | | AP | | |
|---|---|---|---|---|---|---|---|---|---|
| | RIGID | AEROBLADE | Ours | RIGID | AEROBLADE | Ours | RIGID | AEROBLADE | Ours |
| ADM | 0.5144 | 0.5065 | **0.5727** | 0.5715 | 0.5079 | **0.6811** | 0.5635 | 0.5701 | **0.6541** |
| BigGan | 0.5298 | 0.5831 | **0.7756** | 0.5300 | 0.5932 | **0.8575** | 0.5400 | 0.6132 | **0.8680** |
| CRN | 0.5000 | 0.3065 | **0.7302** | 0.0945 | 0.1875 | **0.8637** | 0.3202 | 0.3411 | **0.8597** |
| CycleGan | 0.4640 | 0.5611 | **0.7342** | 0.3760 | 0.5911 | **0.9013** | 0.4119 | 0.5617 | **0.9052** |
| DALL-E | 0.5071 | 0.4016 | **0.7772** | 0.4951 | 0.3852 | **0.8660** | 0.5152 | 0.4104 | **0.8810** |
| GauGan | 0.4804 | 0.5759 | **0.8794** | 0.3181 | 0.6212 | **0.9770** | 0.3924 | 0.6315 | **0.9792** |
| Glide 100 10 | 0.4591 | 0.2939 | **0.8831** | 0.0484 | 0.0588 | **0.9534** | 0.3120 | 0.3142 | **0.9588** |
| Glide 100 27 | 0.4594 | 0.3026 | **0.8713** | 0.0662 | 0.1017 | **0.9462** | 0.3139 | 0.3252 | **0.9533** |
| Glide 50 27 | 0.4591 | 0.3000 | **0.8843** | 0.0471 | 0.0754 | **0.9588** | 0.3114 | 0.3159 | **0.9625** |
| Guided | 0.4803 | 0.5421 | **0.7390** | 0.3513 | 0.5590 | **0.8540** | 0.4135 | 0.5883 | **0.8594** |
| IMLE | 0.5000 | 0.3316 | **0.7311** | 0.0502 | 0.2310 | **0.8699** | 0.3128 | 0.3524 | **0.8763** |
| LDM 100 | 0.4685 | 0.4295 | **0.8232** | 0.3241 | 0.4489 | **0.8781** | 0.3890 | 0.4423 | **0.9096** |
| LDM 200 | 0.4669 | 0.4353 | **0.8268** | 0.3284 | 0.4587 | **0.8758** | 0.3923 | 0.4449 | **0.9081** |
| Midjourney | **0.9407** | 0.4020 | 0.5545 | **0.9890** | 0.3808 | 0.6278 | **0.9906** | 0.4057 | 0.6086 |
| SAN | 0.3746 | 0.3485 | **0.6000** | 0.3516 | 0.3985 | **0.6131** | 0.4359 | 0.4159 | **0.5738** |
| SD v1.4 | **0.8696** | 0.5264 | 0.6196 | **0.9555** | 0.5747 | 0.7818 | **0.9535** | 0.5403 | 0.7230 |
| SD v1.5 | **0.8718** | 0.5512 | 0.6300 | **0.9543** | 0.6027 | 0.7955 | **0.9514** | 0.5764 | 0.7339 |
| Stylegan | 0.5335 | 0.6081 | 0.7045 | 0.5932 | 0.6496 | 0.7225 | 0.5770 | 0.6388 | **0.7649** |
| Stylegan2 | 0.4705 | **0.6467** | 0.6422 | 0.4641 | **0.7125** | 0.7100 | 0.4565 | 0.6836 | **0.7275** |
| ProGan | 0.4769 | 0.5306 | **0.9032** | 0.3216 | 0.5493 | **0.9689** | 0.3935 | 0.5450 | **0.9738** |
| VDQM | 0.5217 | 0.5282 | **0.7686** | 0.4852 | 0.5486 | **0.8744** | 0.5245 | 0.5939 | **0.8783** |
| Wukong | **0.8777** | 0.5135 | 0.6539 | **0.9471** | 0.5502 | 0.7935 | **0.9552** | 0.5249 | 0.7662 |
| Average | 0.5557 | 0.4648 | **0.7411** | 0.4392 | 0.4448 | **0.8350** | 0.5194 | 0.4925 | **0.8330** |

## H  LOCAL MAXIMA PREVIOUS OBSERVATIONS

In Fig. 12 we provide excerpts of other works that include relevant observations regarding the tendency to learn distributions with local maximas.

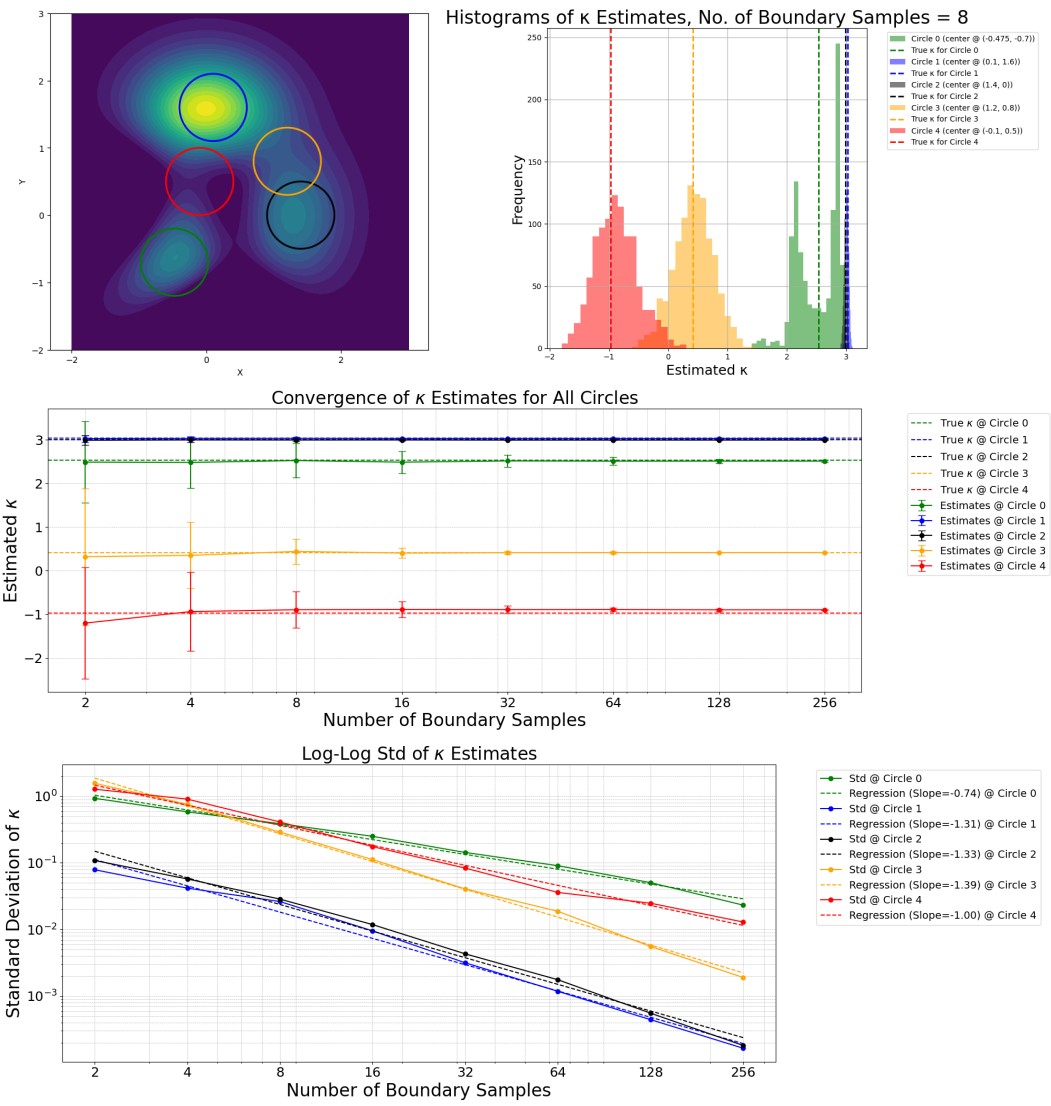

Figure 11: The same experiment as in Fig. 3(b-d), but with all 5 interest points and their corresponding neighborhoods. The circle colors are consistent across all subplots and associated measurements. We still get a consistent estimator of $\kappa$. We also still get good separability between local maximas and saddle points even upon low sample sizes, as shown in the second row, and in more detail in the histograms (top right): These further demonstrate that saddle points (colored red and orange) are distinguishable from local maxima (colored green, blue, and black), despite errors induced by a low number of boundary samples $s = 8$. In other words - good robustness to the hyper-parameter $s$ is observed in terms of distinguishing local maximas from saddle points. This settles well with the robustness to $s$ in terms of detecting generated images, demonstrated in Table 2 and Fig. 10).

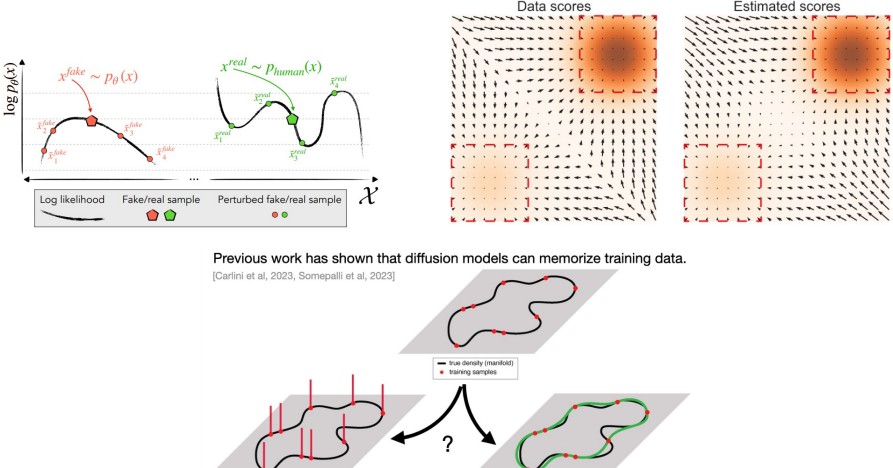

Figure 12: **Top Left:** Excerpt from Mitchell et al. (2023), Fig. 2, showing their hypothesis that generated text lies near local maxima points of the learned probability. **Top Right:** Excerpt from Song & Ermon (2019), Fig. 2. A diffusion model was trained on data from a bi-modal GMM. We can see that the true data probability has 3 basins of attraction: A narrow basin across the diagonal, that leads to a saddle point, and two other basins leading to local maxima. However, the modeled score function learned by a diffusion model predominantly learns the 2 maxima basins, which "overtake" most of the the non-local-maxima basin. **Bottom:** An excerpt from Kadkhodaie et al. (2024) that illustrates how diffusion models may have bias due to training data (in this case, the small size training set causes memorization) - which in turn results in bumps in the implicitly learned probability desnity function.

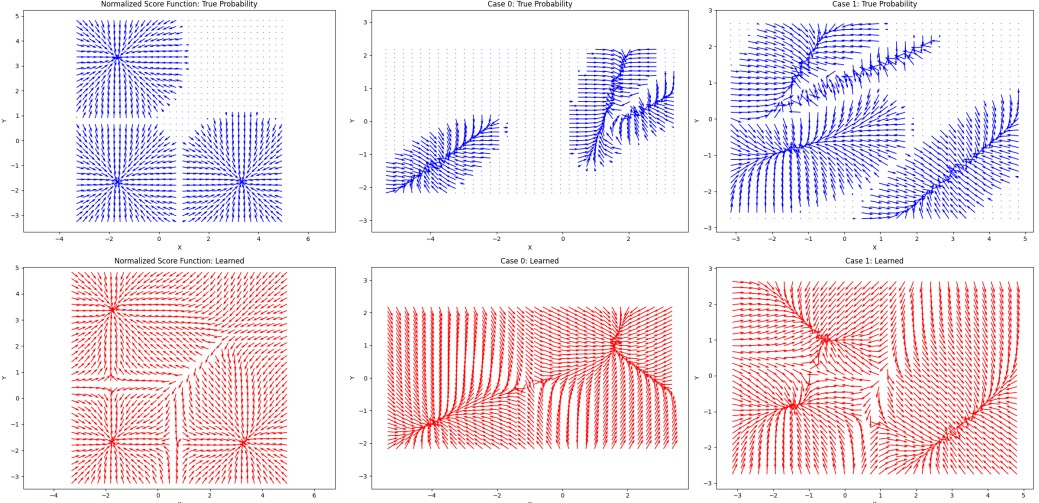

Figure 13: **Score-function fields: True vs Learned**. We experimented with the 2D GMM data of Fig. 3(a) and 2 additional cases, each case with re-drawn hyper-parameters. A diffusion model was trained on the data as in Fig. 3(a). The quiver plots show the normalized score-function as *learned* by the diffusion model, and its ground-truth counterpart obtained from the *true probability*. Clearly, the local maxima basins of attraction in the learned probability "take over" regions where the true field is low. This supports the assumption that the learned reverse diffusion terminates near local maxima. Formally, the field $f$ was normalized as, $f_{\text{normed}}(x) = \frac{f(x)}{\|f(x)\| + 10^{-8}}$ (our method uses the same normalization for the generated image detection criterion). For this visualization, $x$ is sampled uniformly on the X-Y grid.

