# OpenReview forum: "Manifold Induced Biases for Zero-shot and Few-shot Detection of Generated Images"
_ICLR.cc/2025/Conference — ICLR 2025 Poster_

### Official Review · Reviewer_qnVq · 2024-10-23

**Soundness:** 3
**Presentation:** 3
**Contribution:** 3
**Rating:** 6
**Confidence:** 3

**Summary:**

The paper presents a novel approach to detecting AI-generated images using a zero-shot and few-shot framework. By analyzing biases inherent in the manifold of pre-trained diffusion models, the authors introduce a new mathematical criterion based on score functions, curvature, and gradient analysis. This approach generalizes well to unseen generative techniques and outperforms existing methods in both zero-shot and few-shot scenarios. Extensive experiments across a diverse set of generative models further validate the effectiveness of the proposed method.

**Strengths:**

The paper provides a sound theoretical foundation by integrating manifold analysis with diffusion models, advancing the field of generated image detection.

The empirical results show strong performance, with the proposed method outperforming current state-of-the-art approaches.

The authors conducted experiments across various datasets and generative techniques, including GANs, diffusion models, and commercial tools, providing strong evidence for the robustness of the method.

**Weaknesses:**

According to the article, the proposed curvature and gradient based metric for detecting generated images is closely related to the score function. However, it is unclear why this method also performs well with models where score functions are not inherently applicable, such as CycleGAN. The authors are encouraged to clarify this connection and explain why the proposed metric shows strong performance even in such cases where score function analysis is not directly relevant.

There are citation issues on page 7 of the article, where footnotes and page numbers are not correctly referenced. The authors should revise the citation formatting to ensure all references are accurate and properly aligned with the content.

The proposed method appears to fit naturally within a zero-shot framework, relying solely on input samples and corresponding perturbations. It is unclear why the few-shot setting was introduced, given that zero-shot scenarios are typically more challenging and often reflect real-world situations. The authors are encouraged to clarify the need for a few-shot setting and explain why zero-shot alone would not suffice as a more compelling and realistic approach.

**Questions:**

Please see weakness.

---

> ### Author Response · Authors · 2024-11-19
>
> We  appreciate the recognition of the sound theory and robust performance of our approach. Below we answer the questions raised.
>
> **Q1:** “..it is unclear why this method also performs well with models where score functions are not inherently applicable..”
>
> **A1:**  We appreciate this question, which addresses a fundamental challenge in the field of generative image detection: Mitigating the biases that arise when a method is exposed to specific generative techniques, which could impair its generalization to new, unseen techniques. Although such generalization is a notable strength of our approach, in the Limitations section we acknowledge that we do not have an extensive theory regarding the generalizability to unseen generative techniques.
>
> One possible explanation lies in the shared characteristics of generative model manifolds due to similarity in training data. There is limited availability of large-scale datasets [1,2], and many generative models are trained on similar datasets   , which may lead to overlapping or closely related learned probability manifolds. This is despite the differences in architecture, size, and configuration [3]. Consequently, our method, which is exposed to one model’s manifold, may effectively use it as a proxy for detecting images generated by other generative models. This hypothesis aligns with observations in the literature that models trained on comparable datasets exhibit commonalities [4,5].
>
> An interesting hypothetical challenge in the field may arise with the introduction of an entirely novel dataset—one that no generative model has been trained on to date, followed by training of new generative models. In such a scenario, it is uncertain whether existing methods would maintain their performance on the resulting newly generated images, as their success may be tied to biases in currently available datasets. However, we believe that our method can adapt effectively, and that substituting the SD1.4 model with a diffusion model trained on the new dataset is likely to yield strong performance.
>
> To better clarify this in the manuscript, we expanded the discussion in the Limitations section to better outline the shared data perspective and include the references added here.
>
> [1] Villalobos , P., Ho, A., Sevilla, J., Besiroglu, T., Heim, L., & Hobbhahn, M. Position: Will we run out of data? Limits of LLM scaling based on human-generated data. In Forty-first International Conference on Machine Learning.‏
>
> [2] https://www.bloomberg.com/news/articles/2024-11-13/openai-google-and-anthropic-are-struggling-to-build-more-advanced-ai?embedded-checkout=true
>
> [3] Nalisnick, E., Matsukawa, A., Teh, Y. W., Gorur, D., & Lakshminarayanan, B. Do Deep Generative Models Know What They Don't Know?. In International Conference on Learning Representations.
>
> [4] Kornblith, S., Norouzi, M., Lee, H., & Hinton, G. (2019, May). Similarity of neural network representations revisited. In International conference on machine learning (pp. 3519-3529). PMLR.
>
> [5] Nguyen, T., Raghu, M., & Kornblith, S. (2020). Do wide and deep networks learn the same things? uncovering how neural network representations vary with width and depth. arXiv preprint arXiv:2010.15327.
>
> **Q2:** Citation issues on page 7
>
> **A2:** In the revised version, we have corrected all citation formatting and ensured that footnotes and page numbers are accurately referenced.
>
> **Q3:** " The authors are encouraged to clarify the need for a few-shot setting and explain why zero-shot alone would not suffice as a more compelling and realistic approach."
>
> **A3:** We agree that zero-shot scenarios often reflect real-world situations, and our solution is fully suitable for such cases.
>
>
> Nonetheless, in some scenarios, managing a small amount of generated data can be justified if it leads to significant performance improvements - making few-shot methods appropriate. Our research demonstrates that integrating our method with a SOTA few-shot technique, Cozzolino et al. (2024), yields a 4–7% improvement in detection performance without violating the few-shot setting.
>
> For users willing to invest in data maintenance, our method provides a valuable enhancement to few-shot frameworks. It serves as an easy-to-integrate plugin, offering a flexible trade-off between data availability and performance improvement.
>
> Even though few-shot methods for generated image detection offer practical potential, balancing a trade-off between maintenance and performance, this domain is still largely under-explored (as are zero-shot methods). Our work contributes a practical approach to both regimes, excelling in zero-shot scenarios and enhancing few-shot performance.
>
> We have clarified the motivation for introducing the few-shot evaluation in the revised manuscript in Section 5.2, under *Mixture of Experts with Few-shot Approaches*.

---

> > ### Comment · Reviewer_qnVq · 2024-12-01
> > **Thanks for reply**
> >
> > Overall, my concern has been addressed. However, the claim that 'One possible explanation lies in the shared characteristics of generative model manifolds due to similarity in training data' needs further exploration and proof. Therefore, I lean toward maintaining my score.

---

> > > ### Author Response · Authors · 2024-12-01
> > >
> > > Thank you for the feedback, and for the recognition of our contributions to the field.

---

### Official Review · Reviewer_GHKQ · 2024-10-31

**Soundness:** 4
**Presentation:** 3
**Contribution:** 3
**Rating:** 6
**Confidence:** 2

**Summary:**

This work addresses the challenge of distinguishing between real and AI-generated images by analyzing biases on the probability manifold of pre-trained diffusion models. The authors develop a method that offers a scalar criterion for classification in zero-shot settings, and experiments demonstrate its effectiveness against current methods.

**Strengths:**

The theoretical analysis of current diffusion models’ score functions is comprehensive and novel, potentially inspiring further research in this area.

The proposed method generalizes well to unseen generative techniques and achieves superior performance over existing approaches in both zero-shot and few-shot settings.

**Weaknesses:**

The implementation details in Section 4.3 should be elaborated further to enhance readability and reproducibility.


Typo errors:
- L153 and L149, inconsistent use of $\mathcal N$ and $N$.
- L146 and L170, inconsistent use of $\mathbb{R}$ and $R$
- L157, better use latex log $\log$ for clarity.
- L191, use latex \` for upper quotas.
- inconsistent use of Sec. and Section
- L298-L299, unexpected equation.

**Questions:**

The author choose SD-1.4 to implement the proposed method, have the author tried other diffusion models, especially recently more advanced methods, such as SDXL and SD3. Does this helps improve the performance?

---

> ### Author Response · Authors · 2024-11-19
>
> Thank you for the thoughtful feedback and for the careful attention given to our methodology and mathematical details. Below we provide our answers.
>
> **Q1:** Improve the implementation details in Section 4.3
>
> **A1:** Thank you for raising this issue. To improve the readability of Section 4.3 and the ease of reproducibility, the calculation pipeline of $C(x_0)$ was broken down into three clear steps, referencing Fig. 1 to aid with a high-level illustration. Additionally, the CLIP mappings have been restructured into a two-stage process and are now explained in a less formal, more intuitive manner, reducing the overly mathematical approach that previously hindered clarity. We also separated best practices from insights.
>
> Please do not hesitate to raise further questions regarding unclear points or any additional details that may have been overlooked.
>
> **Q2:** Typos
>
> **A2:** We have fixed the typos  and corrected equation presentations - thank you for finding these.
>
> **Q3:** Will incorporating to the method other diffusion models that are more advanced than SD1.4 improve its performance?
>
> **A3:** In Sec. 5.2 under **Sensitivity and Ablation Analysis** we have readily evaluated our method using two advanced diffusion models: **SD v2 Base** and **Kandinsky 2.1**, where the latter was released in September 2023 - the same month as SDXL. We acknowledge this analysis may have been overlooked as it was only mentioned in the text. In the revised manuscript, we have highlighted this experiment by adding a summary table of the results – see Table 2.
>
> Although SD v2 Base and Kandinsky 2.1 models differ in size and generation techniques, our experiments demonstrated consistent results, with a minor AUC decrease of less than 1%. This consistency highlights the robustness of our method across different diffusion models. With that said, it also demonstrates that more advanced models do not necessarily improve the performance.
>
> If further details or additional experiments on this matter would be helpful, please let us know and we would be happy to provide them.

---

> > ### Comment · Reviewer_GHKQ · 2024-12-02
> >
> > Thank the authors for the response. This addressed most of my concerns, I will maintain my score.

---

> > > ### Author Response · Authors · 2024-12-03
> > >
> > > Thank you for the thoughtful review, we are glad our response addressed most of your concerns.

---

### Official Review · Reviewer_DPAQ · 2024-11-02

**Soundness:** 3
**Presentation:** 3
**Contribution:** 3
**Rating:** 6
**Confidence:** 2

**Summary:**

The paper explores a novel method to detect AI-generated images, focusing on zero-shot and few-shot regimes. It identifies key challenges in the field, such as the need for data upkeep with traditional supervised learning methods and limited theoretical grounding for current approaches. The authors propose a framework based on the implicit biases within the manifold of a pre-trained diffusion model, leveraging score-function analysis to approximate manifold curvature and gradient in the zero-shot setting. They extend the method for few-shot scenarios by incorporating a mixture-of-experts strategy. The proposed method demonstrates enhanced performance across 20 generative models.

**Strengths:**

- The idea of leveraging manifold-induced biases from pre-trained diffusion models to detect generated images is novel and interesting.
- The methodology, essential theoretical formulations, and results are well-articulated, with equations and definitions supporting the approach.
- Experimental results are promising, and Figures 1 and 2 are intuitive and easy to understand.

**Weaknesses:**

The primary concern is the robustness; please see the Questions below.

**Questions:**

1. The proposed method relies on pre-trained diffusion models and manifold analysis, which are implemented in high-dimensional space, potentially increasing computational costs. Could the authors provide an analysis of inference time and memory requirements?

2. The method depends on certain hyperparameters, such as perturbation strength and the number of spherical noises. How robust is the method to these parameters across different models, and what guidelines can be provided for selecting these parameters?

3. The authors tested the impact of JPEG compression on the method and reported a slight performance decrease. How does the method perform with other types of image post-processing, such as augmentation, denoising, and flipping?

---

> ### Author Response · Authors · 2024-11-19
>
> We greatly appreciate the reviewers' recognition of our work and the constructive feedback. This prompted us to conduct additional experiments on robustness (Table 2, Fig. 10) and computational efficiency (Appendix Sec. G.3), strengthening the rigor and validation of our approach. Below we answer the questions in detail:
>
> **A1:** Thank you for pointing this out. We conducted additional experiments to analyze our method inference time and memory requirements. Using a single A100 GPU, we observed a runtime of **2.1 seconds per sample** upon fully parallelized processing of the perturbations.
> For comparison, our primary competitor, AEROBLADE, requires **5.4 seconds per sample** on the same A100 GPU, making our method significantly more computationally efficient in terms of runtime.
> The additional memory required by our method is negligible, as it applies the diffusion model during inference and records only a single scalar value per sample as the final result.
> This analysis has been included in the Appendix (see Section G.3) for further reference.
>
> **A2:** Thank you for highlighting the importance of hyperparameter robustness. We provide sensitivity analysis for the perturbation no. and the spherical noises level hyperparameters in Section 5.2 ("Sensitivity and Ablation Analysis") with further details in Appendix Section G.2.
>
> Regarding the **perturbation no.**, our experiments indicate that increasing the no. of perturbations $s$ improves detection performance. Specifically, testing $s$=4, 8, 16, 32, and 64 resulted in average AUC scores of 0.828, 0.829, 0.830, 0.833, and 0.835, respectively. To address the reviewer’s suggestion, **we also evaluated the robustness to perturbation no. across different models**. These experiments revealed only minimal variations in performance (0.1–0.2%) with increased $s$ - see Fig. 10 in Appendix Section G.2.
>
> We also evaluate our method under varied **spherical noise levels**. In our experiment, increasing the radii by a factor of 10 resulted in a 1.5% performance decrease, while increasing by a factor of 100 led to a 2% decrease.
>
> Please note that for improved clarity and presentation, we have added a table in the main manuscript that summarizes the results of this experiment (see Table 2).
>
> These results show that we remain SOTA even under changes in both hyper-parameters. As guidance for noise strength search we have used $\sqrt{d}$ as a point of reference to the sphere's radius, because of its theoretical relation to $\mathcal{N}(0,I)$ in $d$ dimensions (where $d$ is the denoised signal dimension). We also propose to test with higher $s$ and smaller perturbations, since the trend of our tests indicate that these may obtain improved performance.
>
> **A3:** Thank you for this valuable feedback. Indeed, evaluating our method on various image post-processing options would provide valuable information on the robustness of our method to real-world possibilities. To address this, we follow [1, 2] and test Gaussian blur post-processing.
>
> We conducted experiments using Gaussian blur. Specifically, we applied OpenCV’s Gaussian blurring functionality with two kernel sizes (and the default associated variance): medium blur (Kernel Size = 3) and high blur (Kernel Size = 7). Under medium blur conditions, our method’s accuracy decreased by 1.2%, while under high blur conditions, accuracy reductions were 6.2%.
>
> This analysis has been included in the main manuscript in Section 5.2 ‘Sensitivity and Ablation Analysis’ part and in the Appendix (see Section G.2) for further reference.
>
> [1] Ricker, J., Lukovnikov, D., & Fischer, A. (2024). AEROBLADE: Training-Free Detection of Latent Diffusion Images Using Autoencoder Reconstruction Error. In Proceedings of the IEEE/CVF Conference on Computer Vision and Pattern Recognition (pp. 9130-9140).‏
>
> [2] He, Z., Chen, P. Y., & Ho, T. Y. (2024). RIGID: A Training-free and Model-Agnostic Framework for Robust AI-Generated Image Detection. arXiv preprint arXiv:2405.20112.‏

---

> > ### Author Response · Authors · 2024-11-25
> > **Additional Robustness Points**
> >
> > **Point 1: Figure 10 (right panel).** The Appendix now includes the requested per-model perturbation-strength ($\alpha$) sensitivity analysis (in addition to the ($s$) per-model sensitivity analysis). Reminder: Table 2 shows that $\alpha$ can be scaled $\times 100$ while keeping overall SOTA performance. The per-model analysis reveals excellent robustness as well, where even upon $\times 100$ scaling of $\alpha$, most models show less than -0.05 decrease in AUC.
> >
> > **Point 2: Remarkable conclusions regarding the robustness** to the perturbation number ($s$) were drawn from **Figure 3(b–d)**, bridging the practical sensitivity analysis with theory-verifying experiments. Although Figure 3(b–d) was initially conducted in response to Reviewer Xh3J's concerns without an explicit intention to test robustness, this attribute emerged naturally in this setting. We conducted an error analysis of a **2D known, analytic probability function**. In this analysis, the curvature $\kappa$ around a data point is approximated using spherical perturbations according to the formula we employ for image detection. We leverage access to the analytic probability function to calculate the error of this approximation.
> >
> > In **Figure 3(c)**, we observe that the error increases as $s$ decreases but remains sufficiently small to effectively distinguish maxima from saddle points, even at low $s$. Importantly, **distinguishing local maxima on the image probability manifold is our main task**, reminder: We hypothesize that the ability to distinguish maxima translates to the ability to identify generated images. Thus, the robustness to low $s$ in terms of distinguishing maxima in **Figure 3(c)** aligns with the robustness to low $s$ in terms of generated image detection, as detailed in **Table 2 (top right)** and **Figure 10 (left)**.
> >
> > To further emphasize this robustness perspective, we have added per-point histograms, shown in **Figure 11 (top right)** - page 25 of the Appendix, alongside new captions marked in green.

---

### Official Review · Reviewer_Xh3J · 2024-11-03

**Soundness:** 3
**Presentation:** 2
**Contribution:** 3
**Rating:** 8
**Confidence:** 3

**Summary:**

This paper introduces a zero/few-shot framework for detecting AI-generated images by analyzing the inherent biases of a pre-trained diffusion model. The authors hypothesize that generated images are more likely to occupy stable local maxima on this learned manifold, characterized by specific curvature and gradient properties. By approximating these properties, they create a criterion to distinguish between real and generated images without requiring large datasets or retraining. Empirical results show their method outperforms other detection approaches across multiple generative models.

**Strengths:**

- Zero-shot and few-shot capability makes the method practical.
- The theoretical perspective is interesting.
- Empirical results seem promising.

**Weaknesses:**

1. Some theoretical assertions, such as the assumption that generated samples are more likely to be stable local maxima on the learned manifold, are not fully justified. This assumption underpins the detection criterion, but the paper does not offer a thorough mathematical or empirical rationale to support it.
2. The paper relies heavily on approximations in score-function and curvature estimations (e.g., Eq.5, 16-18). However, there is limited discussion or analysis of the tightness of these approximations. This could lead to questions about the reliability of the approximations, especially when they form the foundation of the theoretical claims. It would be beneficial if the authors provided error bounds analysis or empirical justifications for these approximations.
3. In line 122, the authors argue that previous methods still rely on access to generative methods during training, leading to biases towards those generation techniques. However, the proposed approach also relies on a pre-trained SD1.4. How does the proposed approach avoid the bias from it? For example, a realistic sample generated from a more recent model may not sit on a stable local maximum in the learnt log probability manifold of SD1.4.
4. The presentation can be improved. The mathematical analysis can benefit from illustrative examples, while the detailed proof can be moved to appendix.

**Questions:**

See weaknesses.

---

> ### Author Response · Authors · 2024-11-19
>
> Thank you for your constructive feedback, which has been instrumental in improving our work. In response, we empirically demonstrate our key assumptions (Figs. 3.a, 4.c and 7) and test the reliability of our approximations (Fig 3.b-d) through analyzeable and illustrative cases.
>
> **A1:** Thank  you for raising the point on “region of local maxima assumption”. To this end we conducted an experiment with a 2D Gaussian Mixture Model (GMM) dataset - see Appendix B for details. This setting allows for tractable Probability Density Function analysis and was inspired by [1] which also used 2D GMMs to gain insights into diffusion models. As shown in Fig. 3.a, reverse diffusion processes consistently terminate near local maxima, supporting our assumption. This 2D experiment offers intuitive, tractable evidence for similar behaviors in high-dimensional models. Further statistics at larger scale support this in Fig.7 in the Appendix.
>
> Furthermore, this assumption aligns with prior observations: Fig. 2  in [2] illustrates a similar assumption for text generation, albeit relying on LLM’s explicit probability modeling. Fig. 2 in [1]  shows how a diffusion model learns basins of local maxima, which "overtake" a non-maxima basin present in the true probability distribution. Recently, [3] associated suboptimally trained diffusion models with "bumpy" probability manifolds.
>
> Figs. 3.a, 7 support our specific perspective of the local maxima property. Please let us know of additional tests that may further substantiate it.
>
> **A2  :** Thank you for the suggestion to provide an error analysis of the curvature approximations. To this end, we conducted an experiment using an analytic 2D function with complex yet “nice” (differentiable) topography, enabling true quantities for validation.
>
> The curvature $\kappa$ (Eq. 8) around a data point is approximated through discrete sampling of its neighborhood boundary. Though grounded in Gauss Divergence Thm., we recognize that the finite sample size warrants error analysis. The findings, presented in Fig. 3 (b-d), demonstrate the following:
>
> 1. **Expected Behaviour:** The Local maxima receives higher true $\kappa$ than the saddle point (Fig. 3.b).
>
> 2. **Robustness to Low-Sample Approximations:** Even at low sample sizes with large error bars, maxima and saddle points are separable by a threshold (Fig. 3.c). **This separability is the focus of our paper.**
>
> 3. **Consistent Estimator:** The $\kappa$ estimates’ mean is close to the true value across sample sizes (Fig. 3.c), while their std decays exponentially (Fig. 3.d).
>
> This experiment provides essential reliability verification of our theoretical framework. If any important statistics can be added - please inform us. Details are in Appendix A.
>
> **A3  :** We would like to kindly draw your attention to the fact that the bias mentioned in line  122 refers to prior methods that train detectors on generated images. Such training introduces significant bias to the generation techniques encountered during training as documented in Fig. 2 of [4] and Fig. 2 of [5]. In contrast, our zero-shot method  is training-free leveraging a pre-trained SD1.4. Importantly, SD1.4 was pre-trained on real images (without generated images).
>
> We acknowledge the bias potential inherent to using a specific model (SD1.4). To test our method in this regard, we evaluated our approach on over 100K generated and 100K real images  across 20 generation techniques – some introduced after SD1.4. Our approach shows strong generalization to unseen generation techniques (Fig. 5, Table 1), providing thorough evidence that it is not confined to the biases of SD1.4.
>
>
> **A4 :** Another important concept that    requires an illustration is **the concentration of measure**, which describes how high-dimensional Gaussian samples concentrate around a sphere - a phenomenon critical to our derivations. To this end we added an additional 2D illustration in Fig 4.c as follows:
>
> 1. For a $d$-dimensional $\epsilon\sim\mathcal{N}(0,I)$ we obtained the $\chi$-distributed $E\|\epsilon\|$, $\mathrm{Var}(\|\epsilon\|)$ (norm’s mean and variance).
>
> 2. We visualized corresponding 2D samples scaled to have these mean and variance of their norm, effectively simulating the phenomenon in 2D.
>
> As $d$ increases, the radius becomes larger, and the variance converges, empirically illustrating the expected “*thin shell*” spherical distribution.
>
> **References**
>
> [1] Song & Ermon, NeurIPS 2019, "Generative modeling by estimating gradients of the data distribution"
>
> [2] Mitchell, et al., ICML 2023, "Detectgpt: Zero-shot machine-generated text detection using probability curvature"
>
> [3] Zahra Kadkhodaie et. al, ICLR 2024, “Generalization in diffusion models arises from geometry-adaptive harmonic representations”
>
> [4] Epstein et. al, ICCV 2023 “Online detection of ai-generated images”
>
> [5] Ojha et. al., CVPR 2023, “Towards universal fake image detectors that generalize across generative models”

---

> > ### Author Response · Authors · 2024-11-19
> > **small update**
> >
> > Two additional figures have been added to the end of the Appendix of the revised manuscript to enhance completeness and clarity:
> >
> > - **Fig. 12, relating to A1**: Presents excerpts from the mentioned prior work's observations, relating to the "region of local maxima" property, providing additional contextual support for our assumptions.
> > - **Fig. 11, relating to A2**: Extends the analysis of Fig. 3 (b-d), verifying the conclusions across all five interest points of the probability function (3 local-maximas, 2 saddle-points).

---

> ### Comment · Reviewer_Xh3J · 2024-11-30
>
> Thanks for the detailed clarification. I have checked the revised materials. As my concerns are addressed, I will raise my rating accordingly.

---

> > ### Author Response · Authors · 2024-11-30
> >
> > Thank you for your feedback, and updated rating.

---

### Author Response · Authors · 2024-11-28
**Discussion Summary**

Dear Reviewers,

We sincerely appreciate your thoughtful feedback, which significantly enhanced our work. As we enter the final 5 days of discussions, we summarize the progress made in addressing all the concerns raised - which led to enhanced clarity, validated approximations, and demonstrated robustness, supported by key experiments presented in Table 2 and Figures 3, 4.c, 7, 10, and 11 (detailed below). We welcome any further feedback or questions you may have.

Best regards,

The Authors

**Summary of Our Contributions and Strengths:**

- **Innovative Theoretical Framework:** We present novel mathematical derivations to detect generated images as points near local maxima of the (log) probability manifold. To this end we present novel derivations, combining score-function analysis and high-dimensional considerations, for curvature and gradient approximations. The resulting formula can be applied with any pre-trained diffusion model.
- **Exceptional Generalization with a +39.1% Average AUC Improvement** over the state-of-the-art zero-shot methods, AEROBLADE and RIGID (2024). This was validated on a comprehensive benchmark of **200K images**, including real and generated samples from **20 diverse generative techniques** (Table 1, Fig. 5). The remarkable improvement stems from our method's superior cross-technique stability, as detailed in Fig. 8 of the Appendix. Furthermore, when integrated into a few-shot setting, our approach enhances performance by **+4% to +7%**, underscoring its applicability.

**Summary of the Key Concerns Raised and our Responses and Experiments:**
- **Justification for the Local Maxima Assumption (Reviewer Xh3J):**
1.  We build on DetectGPT by Mitchell et al., 2023 [1], which revolutionized zero-shot detection of *generated text*, relying on the same assumption (lines 69, 186 in $\textcolor{green}{\text{green}}$). Our research can be perceived as the first extension of [1] to the image domain. However, [1] uses LLM's explicit probability modeling, rendering the method unsuitable for images, since GANs and diffusion models are implicit models. To address this challenge, we base our method on score function analysis.
2. **Fig. 3.a** further justifies this assumption with empirical evidence of reverse diffusion trajectories terminating near local maxima. We used Gaussian Mixture Model data, inspired by Song & Ermon, 2019 [2] (Details and statistics are in Appendix A, Fig. 7).
- **Error Analysis of Curvature Approximations (Reviewer Xh3J):** In **Figs. 3.b–d and 11**, we validate our curvature estimator using an analytic function with ground-truth curvature, demonstrating high reliability:
1. *Expected Behavior:* Local maxima have higher values than saddle points.
2. *Robustness:* Maxima and saddle points are separable even with low sample sizes (Fig. 3.c).
3. *Consistent Estimator:* Our curvature estimations are **empirically unbiased** (Fig. 3.c) with **exponentially decaying std** (Fig. 3.d).

- **Robustness Analysis (Reviewers DPAQ, GHKQ):** **Table 2** presents sensitivity and ablation studies, demonstrating strong robustness, maintaining SOTA performance under all tests requested by the reviewers:
1. Hyperparameter changes.
2. Image corruptions
3. Alternative, more advanced, base diffusion models.
4. **Fig. 10** in the Appendix further details per-model hyperparameter robustness.

- **Visualization and Improved Writing (Reviewers qnVq, Xh3J, GHKQ):**
1. **Fig. 4.c** illustrates the high-dimensional *concentration of measure*, central to our derivations.
2. Writing was clarified, including explanations to calculate our criterion (Sec. 4.3), motivation for the few-shot approach (Sec. 5.2), citations in Limitations section, and typographical corrections.

Thank you once again for your time and detailed feedback. It truly helped us improve our work.

**References (Edited)**

[1] Eric Mitchell, Yoonho Lee, Alexander Khazatsky, Christopher D. Manning, and Chelsea Finn. DetectGPT: Zero-shot machine-generated text detection using probability curvature. In *International Conference on Machine Learning*, pp. 24950–24962. PMLR, 2023.

[2] Yang Song and Stefano Ermon. Generative modeling by estimating gradients of the data distribution. *Advances in Neural Information Processing Systems*, 32, 2019.

**Original comment (references inadvertently omitted):** 28.11

**Edit to include references:** 3.12

---

### Meta-Review · Area_Chair_x2qK · 2024-12-22

**Metareview:**

The paper presents a novel mathematical framework for quantifying bias in generated images. The key idea is that in generated images, there is bias that can distinguish them from real images. All reviewers are positive of the value of the work, its technical novelty and its strong empirical justification.

**Additional Comments On Reviewer Discussion:**

There were no significant comments or changes during the reviewer discussion.

---

### Decision · Program_Chairs · 2025-01-22

Accept (Poster)